# Graph Neural Networks are Inherently Good Generalizers: Insights by Bridging GNNs and MLPs

**Chenxiao Yang, Qitian Wu, Jiahua Wang & Junchi Yan**[*]
Department of CSE & MoE Key Lab of Artificial Intelligence, Shanghai Jiao Tong University
{chr26195,echo740,wangjiahua2001,yanjunchi}@sjtu.edu.cn

## Abstract

Graph neural networks (GNNs), as the de-facto model class for representation learning on graphs, are built upon the multi-layer perceptrons (MLP) architecture with additional message passing layers to allow features to flow across nodes. While conventional wisdom commonly attributes the success of GNNs to their advanced expressivity, we conjecture that this is *not* the main cause of GNNs' superiority in node-level prediction tasks. This paper pinpoints the major source of GNNs' performance gain to their intrinsic generalization capability, by introducing an intermediate model class dubbed as P(ropagational)MLP, which is identical to standard MLP in training, but then adopts GNN's architecture in testing. Intriguingly, we observe that PMLPs consistently perform on par with (or even exceed) their GNN counterparts, while being much more efficient in training. Codes are available at https://github.com/chr26195/PMLP.

This finding provides a new perspective for understanding the learning behavior of GNNs, and can be used as an analytic tool for dissecting various GNN-related research problems including expressivity, generalization, over-smoothing and heterophily. As an initial step to analyze PMLP, we show its essential difference to MLP at infinite-width limit lies in the NTK feature map in the post-training stage. Moreover, through extrapolation analysis (i.e., generalization under distribution shifts), we find that though most GNNs and their PMLP counterparts cannot extrapolate non-linear functions for extreme out-of-distribution data, they have greater potential to generalize to testing data near the training data support as natural advantages of the GNN architecture used for inference.

## 1 Introduction

In the past decades, *Neural Networks* (NNs) have achieved great success in many areas. As a classic NN architecture, *Multi-Layer Perceptrons* (MLPs) (Rumelhart et al., 1986) stack multiple *Feed-Forward* (FF) layers with nonlinearity to universally approximate functions. Later, *Graph Neural Networks* (GNNs) (Scarselli et al., 2008b; Bruna et al., 2014; Gilmer et al., 2017; Kipf & Welling, 2017; Veličković et al., 2017; Hamilton et al., 2017; Klicpera et al., 2019; Wu et al., 2019) build themselves upon the MLP architecture, e.g., by inserting additional *Message Passing* (MP) operations amid FF layers (Kipf & Welling, 2017) to accommodate the interdependence between instance pairs.

Two cornerstone concepts lying in the basis of deep learning research are model's *representation* and *generalization* power. While the former is concerned with what function class NNs can approximate and to what extent they can minimize the *empirical risk* $\widehat{\mathcal{R}}(\cdot)$, the latter instead focuses on the inductive bias in the learning procedure, asking how well the learned function can generalize to unseen in- and out-of-distribution samples, reflected by the *generalization gap* $\mathcal{R}(\cdot) - \widehat{\mathcal{R}}(\cdot)$. There exist a number of works trying to dissect GNNs' representational power (e.g., Scarselli et al. (2008a); Xu et al. (2018a); Maron et al. (2019); Oono & Suzuki (2019)), while their generalizability and connections with MLP are far less well-understood.

[*]Correspondence author is Junchi Yan who is also affiliated with Shanghai AI Laboratory. The work was in part supported by National Key Research and Development Program of China (2020AAA0107600), NSFC (62222607), and STCSM (22511105100).

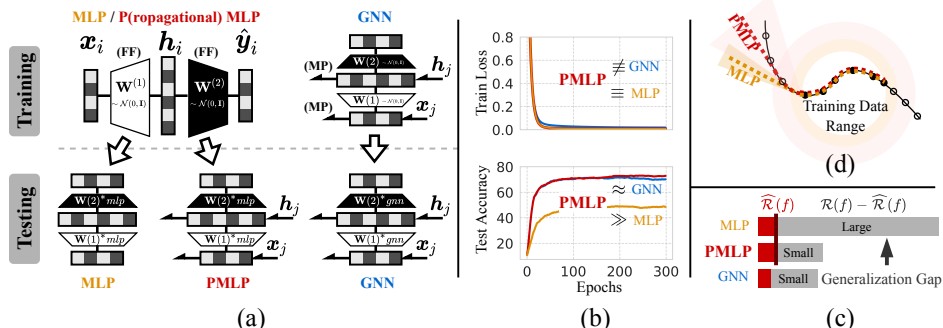

Figure 1: *(a) Model illustration* for MLP, GNN (in GCN-style) and PMLP. *(b) Learning curves* for node classification on Cora that depicts a typical empirical phenomenon. *(c) Intrinsic generalizability* of GNN reflected by close generalization performance of GNN and PMLP. *(d) Extrapolation illustration*: both MLP and PMLP linearize outside the training data support (●: train sample, ○: test sample), while PMLP transits more smoothly and exhibits larger tolerance for OoD testing sample.

In this work, we bridge GNNs and MLPs by introducing an intermediate model class called *Propagational MLPs* (PMLPs). During training, PMLPs are exactly the same as a standard MLP (e.g., same architecture, data for training, initialization, loss function, optimization algorithm). In the testing phase, PMLPs additionally insert non-parametric MP layers amid FF layers, as shown in Fig. 1(a), to align with various GNN architectures including (but not limited in) GCN (Kipf & Welling, 2017), SGC (Wu et al., 2019) and APPNP (Klicpera et al., 2019).

**(Empirical Results and Implications)** According to experiments across sixteen node classification benchmarks and additional discussions on different architectural choices (i.e., layer number, hidden size), model instantiations (i.e., FF/MP layer implementation) and data characteristics (i.e., data split, amount of structural information), we identify two-fold intriguing empirical phenomenons:

● **Phenomenon 1: PMLP significantly outperforms MLP.** Despite that PMLP shares the same weights (i.e., trained model parameters) with a vanilla MLP, it tends to yield lower generalization gap and thereby outperforms MLP by a large margin in testing, as illustrated in Fig. 1(c) and (b) respectively. *This observation suggests that the message passing / graph convolution modules in GNNs can inherently improve model's generalization capability for handling unseen samples.*

The word "inherently" underlines that such particular generalization effects are implicit in the GNN architectures (with message passing mechanism) used in inference, but isolated from factors in the training process, such as: larger hypothesis space for representing a rich set of "graph-aware" functions (Scarselli et al., 2008a; Xu et al., 2018a), more suitable inductive biases in model selection that prioritize those functions capable of relational reasoning (Battaglia et al., 2018), etc.

● **Phenomenon 2: PMLP performs on par with or even exceed GNNs.** PMLP achieves close testing performance to its GNN counterpart in inductive node classification tasks, and can even outperform GNN by a large margin in some cases (i.e., removing self-loops and adding noisy edges). Given that the only difference between GNN and PMLP is the model architecture used in training and the representation power of PMLP is exactly the same with MLP before testing, *this observation suggests that the major (but not only) source of performance improvement of GNNs over MLP in node classification stems from the aforementioned inherent generalization capability of GNNs.*

**(Practical Significance)** We also highlight that PMLP, as a novel class of models (using MLP architecture in training and GNN architecture in inference), can be used for broader analysis purpose or applied as a simple, flexible and very efficient graph encoder model for scalable training.

◇ **PMLP as an analytic tool.** PMLPs can be used for dissecting various GNN-related problems such as over-smoothing and heterophily (see Sec. 3.3 for preliminary explorations), and in a broader sense can potentially bridge theoretical research in two areas by enabling us to conveniently leverage well-established theoretical frameworks for MLPs to enrich those for GNNs.

◇ **PMLP as efficient graph encoders.** While being as effective as GNNs in many cases, PMLPs are *significantly more efficient* in training ($5 \sim 17\times$ faster on large datasets, and $65\times$ faster for very deep GNNs with more than $100$ MP layers). In fact, PMLPs are equivalent to GNNs with all edges dropped in training, which itself (Rong et al., 2020) is a widely recognized way (i.e., DropEdge) for accelerating GNN training. Moreover, PMLPs are more robust against noisy edges, can be trivially

combined with mini-batch training (and many other training tricks for general NNs), and help to quickly evaluate GNN architectures to facilitate model development. Notably, PMLPs can further be extended to transductive learning setting, and are compatible with many other GNN architectures with residual connections (e.g., GCNII (Chen et al., 2020b)) or parametric message passing layers (e.g., GAT (Veličković et al., 2017)) with slight modifications as will be specified.

**(Theoretical Results and Contributions)** As mentioned above, our empirical finding narrows down many different factors between MLPs and GNNs to a key one that attributes to their performance gap, i.e., improvement in generalizability due to the change in network architecture. Then, a natural question arises: "Why this is the case and how does the GNN architecture (in testing) helps the model to generalize?". We take an initial step towards answering this question:

○ **Comparison of three classes of models in NTK regime.** We compare MLP, PMLP, and GNN in the *Neural Tangent Kernel* (NTK) regime (Jacot et al., 2018), where models are over-parameterized and gradient descent finds the global optima. From this perspective, the distinction of PMLP and MLP is rooted in the change of NTK *feature map* determined by model architecture while fixing their minimum RKHS-norm solutions (Proposition 1). For deeper investigation, we first extend the definition of *Graph Neural Tangent Kernel* (GNTK) (Du et al., 2019) to the node regression setting (Lemma 2), and derive the explicit formula for computing the feature map for PMLP/GNN.

○ **OoD generalization / Extrapolation analysis for PMLPs and GNNs.** We consider an important (yet overlooked) aspect of generalization analysis, i.e., *extrapolation* for Out-of-Distribution (OoD) testing samples (Xu et al., 2021) where testing node features become increasingly outside the training data support. Particularly, we reveal that alike MLP, both PMLP and GNN eventually converge to linear functions when testing samples are infinitely far away from the training data support (Theorem 4). Nevertheless, their convergence rates are smaller than that of MLP by a factor related to node degrees and features' cosine similarity (Theorem 5), which indicates both PMLP and GNN are more tolerant to OoD samples and thus have larger potential to generalize near the training data support (which is often the real-world case). We provide an illustration in Fig. 1(d).

## 1.1 RELATED WORKS

Generalization, especially for feed-forward NNs (i.e., MLPs), has been extensively studied in the general ML field (Arora et al., 2019a; Allen-Zhu et al., 2019; Cao & Gu, 2019). However for GNNs, the large body of existing theoretical works focus on their representational power(e.g., Scarselli et al. (2008a); Xu et al. (2018a); Maron et al. (2019); Oono & Suzuki (2019)), while their generalization capability is less well-understood. For node-level prediction setting, those works in generalization analysis (Scarselli et al., 2018; Verma & Zhang, 2019; Baranwal et al., 2021; Ma et al., 2021) mainly aim to derive generalization bounds, but did not establish connections with MLPs since they assume the same GNN architecture in training and testing. For theoretical analysis, the most relevant work is (Xu et al., 2021) that studies the extrapolation behavior of MLP. Their results will later be used in this work. The authors also shed lights on the extrapolation power of GNNs, but for graph-level prediction with max/min propagation from the perspective of algorithmic alignment, which cannot apply to GNNs with average/sum propagation in node-level prediction that are more commonly used.

Regarding the relation between MLPs and GNNs, there are some recent attempts to boost the performance of MLP to approach that of GNN, by using label propagation (Huang et al., 2021), contrastive learning (Hu et al., 2021), knowledge distillation (Zhang et al., 2022) or additional regularization in training (Zhang et al., 2023). However, it is unclear whether these graph-enhanced MLPs can explain the success of GNNs since it is still an open research question to understand these training techniques themselves. There are also few works probing into similar model architectures as PMLP, e.g. Klicpera et al. (2019), which can generally be seen as special cases of PMLP. A concurrent work (Han et al., 2023) further finds that PMLP with an additional fine-tuning procedure can be used to significantly accelerate GNN training. Moreover, a recent work (Baranwal et al., 2023) also theoretically studies how message passing operations benefit multi-layer networks in node classification tasks, which complements our results.

## 2 BACKGROUND AND MODEL FORMULATION

Assume a graph dataset $\mathcal{G} = (\mathcal{V}, \mathcal{E})$ where the node set $\mathcal{V}$ contains $n$ nodes instances $\{(\boldsymbol{x}_u, y_u)\}_{u \in \mathcal{V}}$, where $\boldsymbol{x}_u \in \mathbb{R}^d$ denotes node features and $y_u$ is the label. Without loss of generality, $y_u$ can be a categorical variable or a continuous one depending on specific prediction tasks (classification or

regression). Instance relations are described by the edge set $\mathcal{E}$ and an associated adjacency matrix $\mathbf{A} \in \{0,1\}^{n \times n}$. In general, the problem is to learn a predictor model with $\hat{y} = f(\boldsymbol{x}; \theta, \mathcal{G}_{\boldsymbol{x}}^k)$ for node-level prediction, where $\mathcal{G}_{\boldsymbol{x}}^k$ denotes the $k$-hop ego-graph around $\boldsymbol{x}$ over $\mathcal{G}$.

**Graph Neural Networks and Multi-Layer Perceptrons.** To probe into the connection between mainstream GNNs and MLP from the architectural view, we re-write the GNN formulation in a general form that explicitly disentangles each layer into two operations, namely a *Message-Passing* (MP) operation and then a *Feed-Forwarding* (FF) operation:

$$(\text{MP}): \quad \tilde{\mathbf{h}}_u^{(l-1)} = \sum\nolimits_{v \in \mathcal{N}_u \cup \{u\}} a_G(u,v) \cdot \mathbf{h}_u^{(l-1)}, \quad (\text{FF}): \quad \mathbf{h}_u^{(l)} = \psi^{(l)}\left(\tilde{\mathbf{h}}_u^{(l-1)}\right), \quad (1)$$

where $\mathcal{N}_u$ is the set of neighbored nodes centered at $u$, $a_{\mathcal{G}}(u,v)$ is the affinity function dependent on graph structure $\mathcal{G}$, $\psi^{(l)}$ denotes a feature transformation mapping at the $l$-th layer, and $\mathbf{h}_u^{(0)} = \boldsymbol{x}_u$ is the initial node feature. For example, in Graph Convolution Network (GCN) (Kipf & Welling, 2017), $a_{\mathcal{G}}(u,v) = \mathbf{A}_{uv}/\sqrt{\tilde{d}_u \tilde{d}_v}$, where $\tilde{d}_u$ denotes the degree of node $u$ (with self-loop), and $\psi^{(l)}$ is a fully-connected layer with non-linearity. For an $L$-layer GNN, the prediction is given by $\hat{y}_u = \psi^{(L)}(\mathbf{h}_u^{(L-1)})$, where $\psi^{(L)}$ is often set as linear transformation for regression tasks or with Softmax for classification tasks. Note that GNN models in forms of Eq. 1 degrade to an MLP with a series of FF layers after removing all the MP operations:

$$\hat{y}_u = \psi^{(L)}(\cdots(\psi^{(1)}(\mathbf{x}_u)) = \psi(\mathbf{x}_u). \quad (2)$$

**Typical Types of GNN Architectures.** Besides GCN, many other mainstream GNN models can be written as the architectural form defined by Eq. 1 whose layer-wise updating rule involves MP and FF operations, e.g., GAT (Veličković et al., 2017) and GraphSAINT (Zeng et al., 2019). Some recently proposed node-level Transformer models such as NodeFormer (Wu et al., 2022) and DIFFormer (Wu et al., 2023) also fall into this category. Furthermore, there are also other types of GNN architecture represented by SGC (Wu et al., 2019) and APPNP (Klicpera et al., 2019) where the former adopts multiple MP operations on the initial node features, and the later stacks a series of MP operations at the end of FF layers. These two classes of GNNs are also widely explored and studied. For example, SIGN (Rossi et al., 2020), S$^2$GC (Zhu & Koniusz, 2021) and GBP (Chen et al., 2020a) follow the SGC-style, and DAGNN (Liu et al., 2020c), AP-GCN (Spinelli et al., 2020) and GPR-GNN (Chien et al., 2020) follow the APPNP-style.

**Bridging GNNs and MLPs: Propagational MLP.** After decoupling the MP and FF operations from GNNs' layer-wise updating, we notice that the unique and critical difference of GNNs and MLP lies in whether to adopt MP (somewhere between the input node features and output prediction). To connect two families, we introduce a new model class, dubbed as *Propagational MLP* (PMLP), which has exactly the same architecture as conventional MLP, namely, the same feed-forwarding network. During the inference/testing stage, PMLP$_{GCN}$ incorporates a message passing layer into each layer's feed-forwarding, PMLP$_{SGC}$ adds multiple MP layers in the first layer, and PMLP$_{APP}$ adds them in the last layer. For clear head-to-head comparison, Table 6 in the appendix summarizes the architecture of these models in training and testing stages.

**Extensions of PMLP.** The proposed PMLP is generic and compatible with many other GNN architectures with some slight modifications. For example, we can extend the definition of PMLPs to GNNs with residual connections such as JKNet (Xu et al., 2018b) and GCNII (Chen et al., 2020b) by removing their message passing modules in training, and correspondingly, PMLP$_{KJNet}$ and PMLP$_{GCNII}$ become MLPs with different residual connections in training, which will be further discussed in the next section. For GNNs whose MP layers are parameterized such as GAT (Veličković et al., 2017), one can additionally fine-tune the PMLP model using the corresponding GNN architecture on top of pre-trained FF layers or training MP layers independently.

## 3 EMPIRICAL EVALUATION

We conduct experiments on a variety of node-level prediction benchmarks. **Section 3.1** shows that the proposed PMLPs can significantly outperform the original MLP though they share the same weights, and approach or even exceed their GNN counterparts. **Section 3.2** shows this phenomenon holds across different experimental settings. **Section 3.3** sheds new insights on some research

Table 1: Mean and STD of testing accuracy on node-level prediction benchmark datasets.

| | Dataset
#Nodes | Cora
2,708 | Citeseer
3,327 | Pubmed
19,717 | A-Photo
7,650 | A-Computer
13,752 | Coauthor-CS
18,333 | Coauthor-Physics
34,493 |
|---|---|---|---|---|---|---|---|---|
| GNNs | GCN | $74.82 \pm 1.09$ | $67.60 \pm 0.96$ | $76.56 \pm 0.85$ | $89.69 \pm 0.87$ | $78.79 \pm 1.62$ | $91.79 \pm 0.35$ | $91.22 \pm 0.18$ |
| | SGC | $73.96 \pm 0.59$ | $67.34 \pm 0.54$ | $76.00 \pm 0.59$ | $83.42 \pm 2.47$ | $77.10 \pm 2.54$ | $91.24 \pm 0.59$ | $89.18 \pm 0.46$ |
| | APPNP | $75.02 \pm 2.17$ | $66.58 \pm 0.77$ | $76.48 \pm 0.49$ | $89.51 \pm 0.86$ | $78.29 \pm 0.55$ | $91.64 \pm 0.34$ | $91.80 \pm 0.77$ |
| MLPs | MLP | $55.30 \pm 0.58$ | $56.20 \pm 1.27$ | $70.76 \pm 0.78$ | $75.61 \pm 0.63$ | $63.07 \pm 1.67$ | $87.51 \pm 0.51$ | $85.09 \pm 4.11$ |
| | **PMLP$_{GCN}$**
$\Delta_{GNN}$
$\Delta_{MLP}$ | $\mathbf{75.86 \pm 0.93}$
$+1.39\%$
$+37.18\%$ | $\mathbf{68.00 \pm 0.70}$
$+0.59\%$
$+21.00\%$ | $\mathbf{76.06 \pm 0.55}$
$-0.65\%$
$+7.49\%$ | $\mathbf{89.10 \pm 0.88}$
$-0.66\%$
$+17.84\%$ | $\mathbf{78.05 \pm 1.21}$
$-0.94\%$
$+23.75\%$ | $\mathbf{91.76 \pm 0.27}$
$-0.03\%$
$+4.86\%$ | $\mathbf{91.35 \pm 0.82}$
$+0.14\%$
$+7.36\%$ |
| | **PMLP$_{SGC}$**
$\Delta_{GNN}$
$\Delta_{MLP}$ | $\mathbf{75.04 \pm 0.95}$
$+1.46\%$
$+35.70\%$ | $\mathbf{67.66 \pm 0.64}$
$+0.48\%$
$+20.39\%$ | $\mathbf{76.02 \pm 0.57}$
$+0.03\%$
$+7.43\%$ | $\mathbf{86.50 \pm 1.40}$
$+3.69\%$
$+14.40\%$ | $\mathbf{74.72 \pm 3.86}$
$-3.09\%$
$+18.47\%$ | $\mathbf{91.09 \pm 0.50}$
$-0.16\%$
$+4.09\%$ | $\mathbf{89.34 \pm 1.40}$
$+0.18\%$
$+4.99\%$ |
| | **PMLP$_{APP}$**
$\Delta_{GNN}$
$\Delta_{MLP}$ | $\mathbf{75.84 \pm 1.36}$
$+1.09\%$
$+37.14\%$ | $\mathbf{67.52 \pm 0.82}$
$+1.41\%$
$+20.14\%$ | $\mathbf{76.30 \pm 1.44}$
$-0.24\%$
$+7.83\%$ | $\mathbf{88.47 \pm 1.64}$
$-1.16\%$
$+17.01\%$ | $\mathbf{78.07 \pm 2.10}$
$-0.28\%$
$+23.78\%$ | $\mathbf{91.64 \pm 0.46}$
$+0.00\%$
$+4.72\%$ | $\mathbf{91.96 \pm 0.51}$
$+0.17\%$
$+8.07\%$ |

Table 2: Mean and STD of testing accuracy on three large-scale datasets.

| | GCN | SGC | APPNP | MLP | **PMLP$_{GCN}$** | **PMLP$_{SGC}$** | **PMLP$_{APP}$** |
|---|---|---|---|---|---|---|---|
| OGBN-Arxiv
$(\Delta_{GNN}/\Delta_{MLP})$ | $69.04 \pm 0.18$ | $68.56 \pm 0.14$ | $69.19 \pm 0.12$ | $53.86 \pm 0.28$ | $\mathbf{63.74 \pm 2.28}$
$(-7.68\%/+18.34\%)$ | $\mathbf{62.65 \pm 0.35}$
$(-8.62\%/+16.32\%)$ | $\mathbf{63.30 \pm 0.17}$
$(-8.51\%/+17.53\%)$ |
| Train loss
Train time | 1.0480
63.48 ms | 1.1124
90.50 ms | 1.0396
87.24 ms | 1.5917
7.78 ms | **1.5917**
**7.78 ms** | **1.5917**
**7.78 ms** | **1.5917**
**7.78 ms** |
| OGBN-Products
$(\Delta_{GNN}/\Delta_{MLP})$ | $71.35 \pm 0.19$ | $71.17 \pm 0.29$ | $70.41 \pm 0.07$ | $56.24 \pm 0.10$ | $\mathbf{69.71 \pm 0.13}$
$(-2.30\%/+23.95\%)$ | $\mathbf{70.09 \pm 0.13}$
$(-1.52\%/+24.63\%)$ | $\mathbf{65.72 \pm 0.09}$
$(-6.66\%/+16.86\%)$ |
| Train loss
Train time | 0.4013
288.61 ms | 0.4018
665.06 ms | 0.4311
527.26 ms | 1.0841
39.73 ms | **1.0841**
**39.73 ms** | **1.0841**
**39.73 ms** | **1.0841**
**39.73 ms** |
| Flickr
$(\Delta_{GNN}/\Delta_{MLP})$ | $49.66 \pm 0.57$ | $50.93 \pm 0.16$ | $45.31 \pm 0.28$ | $46.44 \pm 0.14$ | $\mathbf{49.55 \pm 1.30}$
$(-0.22\%/+6.70\%)$ | $\mathbf{50.99 \pm 0.39}$
$(+0.12\%/+9.80\%)$ | $\mathbf{44.31 \pm 0.24}$
$(-2.21\%/-4.59\%)$ |
| Train loss
Train time | 1.1462
36.82 ms | 1.4030
50.42 ms | 0.7449
163.82 ms | 1.3344
7.44 ms | **1.3344**
**7.44 ms** | **1.3344**
**7.44 ms** | **1.3344**
**7.44 ms** |

problems around GNNs including over-smoothing, model depth and heterophily. We present the key experimental results in the main text and defer extra results and discussions to Appendix F and G.

We consider sixteen node classification benchmarks involving different types of networks. For fair comparison, we set the layer number and hidden size to the same values for GNN, PMLP and MLP in the same dataset. We basically use GCN convolution for MP layer and ReLU activation for FF layer for all models unless otherwise stated. PMLPs use the MLP architecture for validation. For SGC, we use a MLP instead of one FF layer after linear message passing, and for APPNP, we remove the residual connection (i.e., $\alpha = 0$) such that the MP layer is aligned with other models. More details about implementation, datasets and hyperparameters are deferred to Appendix F.

We adopt inductive learning setting as the evaluation protocol, which is a commonly used benchmark setting by the community, and guarantee a fair comparison between PMLP and its GNN counterpart by ensuring the information of validation/testing nodes is not used in training for all models. Specifically, for node set $\mathcal{V} = \mathcal{V}_{tr} \cup \mathcal{V}_{te}$ where $\mathcal{V}_{tr}$ (resp. $\mathcal{V}_{te}$) denotes training (resp. testing) nodes, the training process is only exposed to $\mathcal{G}_{tr} = \{\mathcal{V}_{tr}, \mathcal{E}_{tr}\}$, where $\mathcal{E}_{tr} \subset \mathcal{V}_{tr} \times \mathcal{V}_{tr}$ only contains edges for nodes in $\mathcal{V}_{tr}$ and the trained model is tested with the whole graph $\mathcal{G}$ for prediction on $\mathcal{V}_{te}$.

## 3.1 MAIN RESULTS

**How do PMLPs perform compared with GNNs and MLP on common benchmarks?** The main results for comparing the testing accuracy of MLP, PMLPs and GNNs on seven benchmark datasets are shown in Table. 1. We found that, intriguingly, three variants of PMLPs consistently outperform MLP by a large margin on all the datasets despite using the same model with the same set of trainable parameters. Moreover, PMLPs are as effective as their GNN counterparts and can even exceed GNNs in some cases. These results suggest two implications. First, the performance improvement brought by GNNs (or more specifically, the MP operation) over MLP may not purely stem from the more advanced representational power, but the generalization ability. Second, the message passing can indeed contribute to better generalization ability of MLP, though it currently remains unclear how it helps MLP to generalize on unseen testing data. We will later try to shed some light on this question via theoretical analysis in Section 4.

**How do PMLPs perform on larger datasets?** We next apply PMLPs to larger graphs which can be harder for extracting informative features from observed data. As shown in Table 2, PMLP still

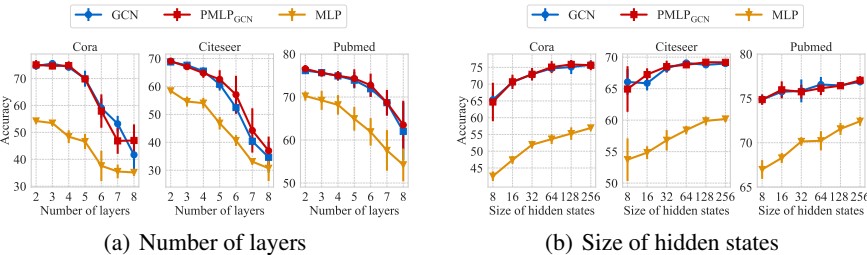

(a) Number of layers           (b) Size of hidden states

Figure 2: Performance variation with increasing layer number and size of hidden states. (See complete results in Appendix G).

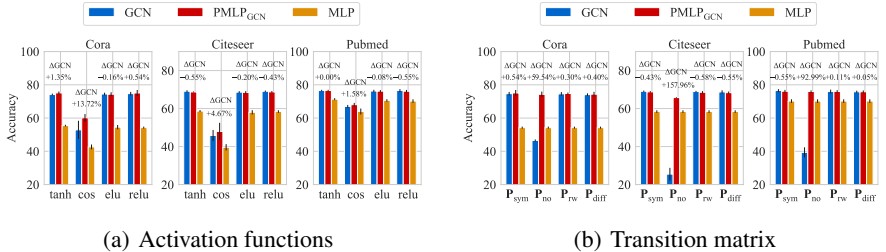

(a) Activation functions           (b) Transition matrix

Figure 3: Performance variation with different activation functions in FF layer and different transition matrices in MP layer. (See complete results in Appendix G)

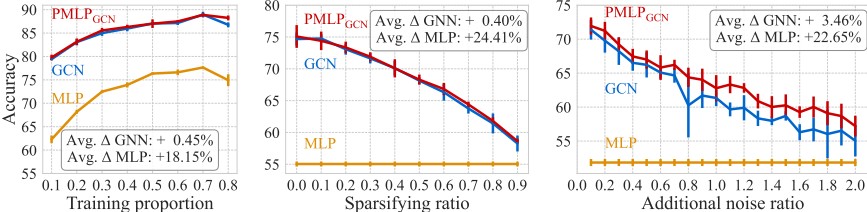

Figure 4: Impact of graph structural information by changing data split, sparsifying the graph, adding random structural noise on Cora. (See complete results in Appendix G)

considerably outperforms MLP. Yet differently, there is a certain gap between PMLP and GNN. We conjecture that this is because in such large graphs the relations between inputs and target labels can be more complex, which requires more expressive architectures for learning desired node-level representations. This hypothesis is further validated by the results of training losses in Table 2, which indeed shows that GNNs can yield lower fitting error on the training data.

## 3.2 FURTHER DISCUSSIONS

We next conduct more experiments and comparison for verifying the consistency of the observed phenomenon across different settings regarding model implementation and graph property. We also try to reveal how PMLPs work for representation learning through visualizations of the produced embeddings, and the results are deferred to Appendix G.

**Q1: What is the impact of model layers and hidden sizes?** In Fig. 2(a) and (b), we plot the testing accuracy of GCN, PMLP$_{GCN}$ and MLP w.r.t. different layer numbers and hidden sizes. The results show that the observed phenomenon in Section 3.1 consistently holds with different settings of model depth and width, which suggests that the generalization effect of the MP operation is insensitive to model architectural hyperparameters. The increase of layer numbers cause performance degradation for all three models, presumably because of over-fitting. We will further discuss the impact of model depth (where layer number exceeds 100) and residual connections in the next subsection.

**Q2: What is the impact of different activation functions in FF layers?** As shown in Fig. 3(a), the relative performance rankings among GCN, PMLP and MLP stay consistent across four different activation functions (tanh, cos, ELU, ReLU) and in particular, in some cases the performance gain of PMLP over GCN is further amplified (e.g., with cos activation).

**Q3: What is the impact of different propagation schemes in MP layers?** We replace the original transition matrix $\mathbf{P}_{\text{sym}} = \tilde{\mathbf{D}}^{-\frac{1}{2}}\tilde{\mathbf{A}}\tilde{\mathbf{D}}^{-\frac{1}{2}}$ used in the MP layer by other commonly used transition matrices: 1) $\mathbf{P}_{\text{no-loop}} = \mathbf{D}^{-\frac{1}{2}}\mathbf{A}\mathbf{D}^{-\frac{1}{2}}$, i.e., removing self-loop; 2) $\mathbf{P}_{\text{rw}} = \tilde{\mathbf{D}}^{-1}\tilde{\mathbf{A}}$, i.e., random walk matrix; 3) $\mathbf{P}_{\text{diff}} = \sum_{k=0}^{\infty} \frac{1}{e \cdot k!}(\tilde{\mathbf{D}}^{-1}\tilde{\mathbf{A}})^k$, i.e., heat kernel diffusion matrix. The results are presented in Fig. 3(b) where we found that the relative performance rankings of three models keep nearly unchanged after replacing the original MP layer. And, intriguingly, the performance of GNNs degrade dramatically after removing the self-loop, while the accuracy of PMLPs stays at almost the same level. The possible reason is that the self-loop connection plays an important role in GCN's training stage for preserving enough centered nodes' information, but does not affect PMLP.

**Q4: What is the impact of training proportion?** We use random split to control the amount of graph structure information used in training. As shown in Fig. 4(left), the labeled portion of nodes and the amount of training edges have negligible impacts on the relative performance of three models.

**Q5: What is the impact of graph sparsity?** As suggested by Fig. 4(middle), when the graph goes sparser, the absolute performance of GCN and PMLP degrades yet their performance gap remains unchanged. This shows that the quality of input graphs indeed impacts the testing performance and tends to control the performance upper bound of the models. Critically though, the generalization effect brought by PMLP is insensitive to the graph completeness.

**Q6: What is the impact of noisy structure?** As shown in Fig. 4(right), the performances of both PMLP and GCN tend to decrease as we gradually add random connections to the graph, whose amount is controlled by the noise ratio (defined as # noisy edges$/|\mathcal{E}|$), while PMLP shows better robustness to such noise. This indicates noisy structures have negative effects on both training and generalization of GNNs and one may use PMLP to mitigate this issue.

### 3.3 Over-Smoothing, Model Depth, and Heterophily

We conduct additional experiments to shed lights on broader aspects of GNNs including over-smoothing, model depth and graph heterophily. Detailed results and respective discussions are deferred to Appendix E. In a nutshell, we find the phenomenon still holds for GNNs with residual connections (i.e., JKNet and GCNII), very deep GNNs (with more than 100 layers) and heterophilic graphs. This indicates that both over-smoothing and heterophily are problems closely related to failure cases of GNN generalization, and can be mitigated by using MP layers that are more suitable for the data or backbone MLP models with better generalization capability, aligning with some previous studies (Battaglia et al., 2018; Cong et al., 2021).

### 3.4 Extension to Transductive Settings

Notably, the current training procedure of PMLP does not involve unlabeled nodes but can be extended to such scenario by combining with existing semi-supervised learning approaches (Van Engelen & Hoos, 2020) such as using label propagation (Zhu et al., 2003) to generate pseudo labels for unlabeled nodes, or additionally using the GNN architecture for fine-tuning, which is still shown to be more efficient than training GNNs from scratch (Han et al., 2023).

## 4 Theoretical Insights on GNN Generalization

Towards theoretically answering why "GNNs are inherently good generalizers" and explaining the superior generalization performance of PMLP and GNN, we next compare MLP, PMLP and GNN from the *Neural Tangent Kernel* (NTK) perspective, derive the formula for computing *Graph Neural Tangent Kernel* (GNTK) in node regression, and then use the results to examine their extrapolation behaviors, i.e., generalization under distribution shifts. Note that our analysis focuses on the model architecture used in inference, and thus the results presented in Sec. 4.2 are applicable for both GNN and PMLP in both inductive and transductive settings.

### 4.1 NTK Perspective on MLP, PMLP and GNN

**Linearization of Neural Networks.** For a neural network $f(\boldsymbol{x}; \boldsymbol{\theta}) : \mathcal{X} \to \mathbb{R}$ with initial parameters $\boldsymbol{\theta}_0$ and a fixed input sample $\boldsymbol{x}$, performing first-order Taylor expansion around $\boldsymbol{\theta}_0$ yields the linearized form of NNs (Lee et al., 2019) as follows:

$$f^{lin}(\boldsymbol{x}; \boldsymbol{\theta}) = f(\boldsymbol{x}; \boldsymbol{\theta}_0) + \nabla_{\boldsymbol{\theta}} f(\boldsymbol{x}; \boldsymbol{\theta}_0)^{\top} (\boldsymbol{\theta} - \boldsymbol{\theta}_0), \tag{3}$$

where the gradient $\nabla_{\boldsymbol{\theta}} f(\boldsymbol{x}; \boldsymbol{\theta}_0)$ could be thought of as a feature map $\phi(\boldsymbol{x}) : \mathcal{X} \to \mathbb{R}^{|\boldsymbol{\theta}|}$, depending on the specific initialization. As such, when $\boldsymbol{\theta}_0$ is initialized by Gaussian distribution with certain scaling and the network width tends to infinity (i.e., $m \to \infty$, where $m$ denotes the layer width), the feature map becomes constant and is determined by the model architecture (e.g., MLP, GNN and CNN), inducing a kernel called NTK (Jacot et al., 2018):

$$\mathrm{NTK}(\boldsymbol{x}_i, \boldsymbol{x}_j) = \phi_{ntk}(\boldsymbol{x}_i)^\top \phi_{ntk}(\boldsymbol{x}_j) = \langle \nabla_{\boldsymbol{\theta}} f(\boldsymbol{x}_i; \boldsymbol{\theta}), \nabla_{\boldsymbol{\theta}} f(\boldsymbol{x}_j; \boldsymbol{\theta}) \rangle \quad (4)$$

**Kernel Regression with NTK.** Recent works (Liu et al., 2020b;a) show that the spectral norm of Hessian matrix in the Taylor series tends to zero with increasing width by $\Theta(1/\sqrt{m})$, and hence the linearization becomes almost exact. Therefore, training an over-parameterized NN using gradient descent with infinitesimal step size is equivalent to kernel regression with NTK (Arora et al., 2019b):

$$f(\boldsymbol{x}; \boldsymbol{w}) = \boldsymbol{w}^\top \phi_{ntk}(\boldsymbol{x}), \quad \mathcal{L}(\boldsymbol{w}) = \frac{1}{2} \sum_{i=1}^n \left( y_i - \boldsymbol{w}^\top \phi_{ntk}(\boldsymbol{x}_i) \right)^2. \quad (5)$$

We next show the equivalence of MLP and PMLP in training at infinite width limit (NTK regime).

**Proposition 1.** *MLP and its corresponding PMLP have the same minimum RKHS-norm NTK kernel regression solution, but differ from that of GNNs, i.e., $\boldsymbol{w}_{mlp}^* = \boldsymbol{w}_{pmlp}^* \neq \boldsymbol{w}_{gnn}^*$.*

**Implications**. From the NTK perspective, stacking additional message passing layers in the testing phase implies transforming the fixed feature map from that of MLP $\phi_{mlp}(\boldsymbol{x})$ to that of GNN $\phi_{gnn}(\boldsymbol{x})$, while fixing $\boldsymbol{w}$. Given $\boldsymbol{w}_{mlp}^* = \boldsymbol{w}_{pmlp}^*$, the superior generalization performance of PMLP (i.e., the key factor of performance surge from MLP to GNN) can be explained by such transformation of feature map from $\phi_{mlp}(\boldsymbol{x})$ to $\phi_{gnn}(\boldsymbol{x})$ in testing:

$$f_{mlp}(\boldsymbol{x}) = \boldsymbol{w}_{mlp}^{*\top} \phi_{mlp}(\boldsymbol{x}), \quad f_{pmlp}(\boldsymbol{x}) = \boldsymbol{w}_{mlp}^{*\top} \phi_{gnn}(\boldsymbol{x}), \quad f_{gnn}(\boldsymbol{x}) = \boldsymbol{w}_{gnn}^{*\top} \phi_{gnn}(\boldsymbol{x}). \quad (6)$$

This perspective simplifies the subsequent theoretical analysis by setting the focus on the difference of feature map, which is determined by the network architecture used in inference, and empirically suggested to be the key factor attributing to superior generalization performance of GNN and PMLP. To step further, we next derive the formula for computing NTK feature map of PMLP and GNN (i.e., $\phi_{gnn}(\boldsymbol{x})$) in node regression tasks.

**GNTK in Node Regression.** Following previous works that analyse shallow and wide NNs (Arora et al., 2019a; Chizat & Bach, 2020; Xu et al., 2021), we focus on a two-layer GNN using average aggregation with self-connection. By extending the original definition of GNTK (Du et al., 2019) from graph-level regression to node-level regression as specified in Appendix. C, we have the following explicit form of $\phi_{gnn}(\boldsymbol{x})$ for a two-layer GNN.

**Lemma 2.** *The explicit form of GNTK feature map for a two-layer GNN $\phi_{gnn}(x)$ with average aggregation and ReLU activation in node regression is*

$$\phi_{gnn}(\boldsymbol{x}_i) = c \sum_{j \in \mathcal{N}_i \cup \{i\}} \left[ \mathbf{X}^\top \boldsymbol{a}_j \cdot \mathbb{I}_+ \left( \boldsymbol{w}^{(k)\top} \mathbf{X}^\top \boldsymbol{a}_j \right), \boldsymbol{w}^{(k)\top} \mathbf{X}^\top \boldsymbol{a}_j \cdot \mathbb{I}_+ \left( \boldsymbol{w}^{(k)\top} \mathbf{X}^\top \boldsymbol{a}_j \right), \dots \right], \quad (7)$$

*where $c = O(\tilde{d}^{-1})$ is a constant proportional to the inverse of node degree, $\mathbf{X} \in \mathbb{R}^{n \times d}$ is node features, $\boldsymbol{a}_i \in \mathbb{R}^n$ denotes adjacency vector of node $i$, $\boldsymbol{w}^{(k)} \sim \mathcal{N}(\boldsymbol{0}, \boldsymbol{I}_d)$ is a random Gaussian vector in $\mathbb{R}^d$, two components in the brackets repeat infinitely many times with $k$ ranging from 1 to $\infty$, and $\mathbb{I}_+$ is an indicator function that outputs 1 if the input is positive otherwise 0.*

### 4.2 MLP V.S. PMLP IN EXTRAPOLATION

As indicated by Proposition 1, the fundamental difference between MLP and PMLP at infinite width limit stems from the difference of feature map in the testing phase. This reduces the problem of explaining the success of GNN to the question that why this change is significant for generalizability.

**Extrapolation Behavior of MLP**. One important aspect of generalization analysis is regarding model's behavior when confronted with OoD testing samples (i.e., testing nodes that are considerably outside the training support), a.k.a. *extrapolation analysis*. A previous study on this direction (Xu et al., 2021) reveal that a standard MLP with ReLU activation quickly converges to a linear function as the testing sample escapes the training support, which is formalized by the following theorem.

**Theorem 3.** *([Xu et al., 2021](#)). Suppose $f_{mlp}(\boldsymbol{x})$ is an infinitely-wide two-layer MLP with ReLU trained by square loss. For any direction $\boldsymbol{v} \in \mathbb{R}^d$ and step size $\Delta t > 0$, let $\boldsymbol{x}_0 = t\boldsymbol{v}$, we have*

$$\left| \frac{\left( f_{mlp}(\boldsymbol{x}_0 + \Delta t\boldsymbol{v}) - f_{mlp}(\boldsymbol{x}_0) \right) / \Delta t}{c_{\boldsymbol{v}}} - 1 \right| = O(\frac{1}{t}). \tag{8}$$

*where $c_{\boldsymbol{v}}$ is a constant linear coefficient. That is, as $t \to \infty$, $f_{mlp}(\boldsymbol{x}_0)$ converges to a linear function.*

The intuition behind this phenomenon is the fact that ReLU MLPs learn piece-wise linear functions with finitely many linear regions and thus eventually becomes linear outside training data support. Now a naturally arising question is how does PMLP compare with MLP regarding extrapolation?

**Extrapolation Behavior of PMLP**. Based on the explicit formula for $\phi_{gnn}(\boldsymbol{x})$ in Lemma 2, we extend the theoretical result of extrapolation analysis from MLP to PMLP. Our first finding is that, as the testing node feature becomes increasingly outside the range of training data, alike MLP, PMLP (as well as GNN) with average aggregation also converges to a linear function, yet the corresponding linear coefficient reflects its ego-graph property rather than being a fixed constant.

**Theorem 4.** *Suppose $f_{pmlp}(\boldsymbol{x})$ is an infinitely-wide two-layer MLP with ReLU activation trained using squared loss, and adds average message passing layer before each feed-forward layer in the testing phase. For any direction $\boldsymbol{v} \in \mathbb{R}^d$ and step size $\Delta t > 0$, let $\boldsymbol{x}_0 = t\boldsymbol{v}$, and as $t \to \infty$, we have*

$$\left( f_{pmlp}(\boldsymbol{x}_0 + \Delta t\boldsymbol{v}) - f_{pmlp}(\boldsymbol{x}_0) \right) / \Delta t \quad \to \quad c_{\boldsymbol{v}} \sum_{i \in \mathcal{N}_0 \cup \{0\}} (\tilde{d} \cdot \tilde{d}_i)^{-1}. \tag{9}$$

*where $c_{\boldsymbol{v}}$ is the same constant as in Theorem 3, $\tilde{d}_0 = \tilde{d}$ is the node degree (with self-connection) of $\boldsymbol{x}_0$, and $\tilde{d}_i$ is the node degree of its neighbors.*

**Remark**. This result also applies to infinitely-wide two-layer GNN with ReLU activation and average aggregation in node regression settings, except the constant $c_{\boldsymbol{v}}$ is different from that of MLP.

**Convergence Comparison**. Though both MLP and PMLP tend to linearize for outlier testing samples, indicating that they have common difficulty to extrapolate non-linear functions, we find that PMLP in general has more freedom to deviate from the convergent linear coefficient, implying smoother transition from in-distribution (non-linear) to out-of-distribution (linear) regime and thus could potentially generalize to out-of-distribution samples near the range of training data.

**Theorem 5.** *Suppose all node features are normalized, and the cosine similarity of node $\boldsymbol{x}_i$ and the average of its neighbors is deonoted as $\alpha_i \in [0, 1]$. Then, the convergence rate for $f_{pmlp}(\boldsymbol{x})$ is*

$$\left| \frac{\left( f_{pmlp}(\boldsymbol{x}_0 + \Delta t\boldsymbol{v}) - f_{pmlp}(\boldsymbol{x}_0) \right) / \Delta t}{c_{\boldsymbol{v}} \sum_{i \in \mathcal{N}_0 \cup \{0\}} (\tilde{d} \cdot \tilde{d}_i)^{-1}} - 1 \right| = O\left( \frac{1 + (\tilde{d}_{max} - 1)\sqrt{1 - \alpha_{min}^2}}{t} \right). \tag{10}$$

*where $\alpha_{min} = \min\{\alpha_i\}_{i \in \mathcal{N}_0 \cup \{0\}} \in [0, 1]$, and $\tilde{d}_{max} \geq 1$ denotes the maximum node degree in the testing node $\boldsymbol{x}_0$'s neighbors (including itself).*

This result indicates larger node degree and feature dissimilarity imply smoother transition and better compatibility with OoD samples. Specifically, when the testing node's degree is 1 (connection to itself), PMLP becomes equivalent to MLP. As reflection in Eq. 10, $\tilde{d}_{max} = 1$, and the bound degrades to that of MLP in Eq. 8. Moreover, when all node features are equal, message passing will become meaningless. Correspondingly, $\tilde{\alpha}_{min} = 1$, and the bound also degrades to that of MLP.

## 5   MORE DISCUSSIONS AND CONCLUSION

We defer more discussions on other sources of performance gap between MLP and GNN, our current limitations and outlooks to Appendix A.

**Conclusion.** In this work, we bridge MLP and GNN by introducing an intermediate model class called PMLP, which is equivalent to MLP in training, but shows significantly better generalization performance after adding unlearned message passing layers in testing and can rival with its GNN counterpart in most cases. This phenomenon is consistent across different datasets and experimental settings. To shed some lights on this phenomenon, we show despite that both MLP and PMLP cannot extrapolate non-linear functions, PMLP converges slower, indicating smoother transition and better tolerance for out-of-distribution samples.

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

## A    MORE DISCUSSIONS

**Other sources of performance gap between MLP and GNN / When PMLP fails?**  Besides the intrinsic generalizability of GNNs that is revealed by the performance gain from MLP to PMLP in this work, we note that there are some other less significant but non-negligible sources that attributes to the performance gap between GNN and MLP in node prediction tasks:

• **Expressiveness:** While our experiments find that GNNs and PMLPs can perform similarly in most cases, showing great advantage over MLPs in generalization, in practice, there still exists a certain gap between their expressiveness, which can be amplified in large datasets and causes certain degrees of performance difference. This is reflected by our experiments on three large-scale datasets. Despite that, we can see from Table 2 that the intrinsic generalizability of GNN (corresponding to $\Delta_{mlp}$) is still the major source of performance gain from MLP.

• **Semi-Supervised / Transductive Learning:** As our default experimental setting, inductive learning ensures that testing samples are unobserved during training and keeps the comparison among models fair. However in practice, the ability to leverage the information of unlabeled nodes in training is a well-known advantage of GNN (but not the advantage of PMLP). Still, PMLP can be used in transductive setting with training techniques as described in Sec. 3.4.

**Current Limitations and Outlooks.**  The result in Theorem 5 provides a bound to show PMLP's better potential in OoD generalization, rather than guaranteeing its superior generalization capability. Explaining when and why PMLPs perform closely to their GNN counterparts also need further investigations. Moreover, following most theoretical works on NTK, we consider the regression task with squared loss for analysis instead of classification. However, as evidences (Janocha & Czarnecki, 2017; Hui & Belkin, 2020) show squared loss can be as competitive as softmax cross-entropy loss, the insights obtained from regression tasks could also adapt to classification tasks.

## B    PROOF FOR PROPOSITION 1

To analyse the extrapolation behavior of PMLP and compare it with MLP, As mentioned in the main text, training an infinitely wide neural network using gradient descent with infinitesimal step size is equivalent to solving kernel regression with the so-called NTK by minimizing the following squared loss function:

$$f(\boldsymbol{x}; \boldsymbol{w}) = \boldsymbol{w}^\top \phi_{ntk}(\boldsymbol{x}), \quad \mathcal{L}(\boldsymbol{w}) = \frac{1}{2} \sum_{i=1}^{n} \left( y_i - \boldsymbol{w}^\top \phi_{ntk}(\boldsymbol{x}_i) \right)^2. \tag{11}$$

Let us now consider an arbitrary minimizer $\boldsymbol{w}^* \in \mathcal{H}$ in NTK's reproducing kernel Hilbert space. The minimizer could be further decomposed as $\boldsymbol{w}^* = \hat{\boldsymbol{w}}^* + \boldsymbol{w}_\perp$, where $\hat{\boldsymbol{w}}^*$ lies in the linear span of feature mappings for training data and $\boldsymbol{w}_\perp$ is orthogonal to $\hat{\boldsymbol{w}}^*$, i.e.,

$$\hat{\boldsymbol{w}}^* = \sum_{i=1}^{n} \lambda_i \cdot \phi_{ntk}(\boldsymbol{x}_i), \quad \langle \hat{\boldsymbol{w}}^*, \boldsymbol{w}_\perp \rangle_{\mathcal{H}} = 0. \tag{12}$$

One observation is that the loss function is unchanged after removing the orthogonal component $\boldsymbol{w}_\perp$:

$$\begin{aligned}
\mathcal{L}(\boldsymbol{w}^*) &= \frac{1}{2} \sum_{i=1}^{n} \left( y_i - \langle \sum_{i=1}^{n} \lambda_i \cdot \phi_{ntk}(\boldsymbol{x}_i) + \boldsymbol{w}_\perp, \phi_{ntk}(\boldsymbol{x}_i) \rangle_{\mathcal{H}} \right)^2 \\
&= \frac{1}{2} \sum_{i=1}^{n} \left( y_i - \langle \sum_{i=1}^{n} \lambda_i \cdot \phi_{ntk}(\boldsymbol{x}_i), \phi_{ntk}(\boldsymbol{x}_i) \rangle_{\mathcal{H}} \right)^2 = \mathcal{L}(\hat{\boldsymbol{w}}^*).
\end{aligned} \tag{13}$$

This indicates that $\hat{\boldsymbol{w}}^*$ is also a minimizer whose $\mathcal{H}$-norm is smaller than that of $\boldsymbol{w}^*$, i.e., $\|\hat{\boldsymbol{w}}^*\|_{\mathcal{H}} \le \|\boldsymbol{w}^*\|_{\mathcal{H}}$. It follows that the minimum $\mathcal{H}$-norm solution for Eq. 11 can be expressed as a linear combination of feature mappings for training data. Therefore, solving Eq. 11 boils down to solving a linear system with coefficients $\boldsymbol{\lambda} = [\lambda_i]_{i=1}^n$. Resultingly, the minimum $\mathcal{H}$-norm solution is

$$\boldsymbol{w}^* = \sum_{i=1}^{n} \left[ y_i \mathbf{K}^{-1} \right]_i \phi_{ntk}(\boldsymbol{x}_i), \tag{14}$$

where $\mathbf{K} \in \mathbb{R}^{n \times n}$ is the kernel matrix for training data. We see that the final solution is only dependent on training data $\{\boldsymbol{x}_i, y_i\}_{i=1}^n$ and the model architecture used in training (since it determines the form of $\phi_{ntk}(\boldsymbol{x}_i)$). It follows immediately that the min-norm NTK kernel regression solution is equivalent for MLP and PMLP (i.e., $\boldsymbol{w}_{mlp}^* = \boldsymbol{w}_{pmlp}^*$) given that they are the same model trained on the same set of data. In contrast, the architecture of GNN is different from that of MLP, implying different form of feature map, and hence they have different solutions in their respective NTK kernel regression problems (i.e., $\boldsymbol{w}_{mlp}^* \neq \boldsymbol{w}_{gnn}^*$).

## C  PROOF FOR LEMMA 3

### C.1  GNTK FOR GRAPH-LEVEL REGRESSION

Graph Neural Tangent Kernel (GNTK) (Du et al., 2019) is a natural extension of NTK to graph neural networks. Originally, the squared loss function is defined for graph-level regression and the kernel function is defined over a pair of graphs:

$$\text{GNTK}(\mathcal{G}_i, \mathcal{G}_j) = \phi_{gnn}(\mathcal{G}_i)^\top \phi_{gnn}(\mathcal{G}_j) = \langle \nabla_{\boldsymbol{\theta}} f(\mathcal{G}_i; \boldsymbol{\theta}), \nabla_{\boldsymbol{\theta}} f(\mathcal{G}_j; \boldsymbol{\theta}) \rangle, \tag{15}$$

$$\mathcal{L}(\boldsymbol{\theta}) = \frac{1}{2} \sum_{i=1}^n (y_i - f(\mathcal{G}_i; \boldsymbol{\theta}))^2, \tag{16}$$

where $f(\mathcal{G}_i; \boldsymbol{\theta})$ yields prediction for a graph such as the property of a molecule. The formula for calculating GNTK is given by the following (where we modify the original notation for clarity and alignment with our definition of GNTK in node-level regression setting):

$$\left[\text{GNTK}^{(0)}(\mathcal{G}_i, \mathcal{G}_j)\right]_{uu'} = \left[\boldsymbol{\Sigma}^{(0)}(\mathcal{G}_i, \mathcal{G}_j)\right]_{uu'} = \boldsymbol{x}_u^\top \boldsymbol{x}_{u'},$$
$$\text{where} \quad \boldsymbol{x}_u \in \mathcal{V}_i, \boldsymbol{x}_{u'} \in \mathcal{V}_j. \tag{17}$$

The message passing operation in each layer corresponds to:

$$\left[\boldsymbol{\Sigma}_{mp}^{(\ell)}(\mathcal{G}_i, \mathcal{G}_j)\right]_{uu'} = c_u c_{u'} \sum_{v \in \mathcal{N}_u \cup \{u\}} \sum_{v' \in \mathcal{N}_{u'} \cup \{u'\}} \left[\boldsymbol{\Sigma}^{(\ell)}(\mathcal{G}_i, \mathcal{G}_j)\right]_{vv'}$$
$$\left[\text{GNTK}_{mp}^{(\ell)}(\mathcal{G}_i, \mathcal{G}_j)\right]_{uu'} = c_u c_{u'} \sum_{v \in \mathcal{N}_u \cup \{u\}} \sum_{v' \in \mathcal{N}_{u'} \cup \{u'\}} \left[\text{GNTK}^{(\ell)}(\mathcal{G}_i, \mathcal{G}_j)\right]_{vv'}, \tag{18}$$

where $c_u$ denotes a scaling factor. The calculation formula of feed-forward operation (from $\text{GNTK}_{mp}^{(\ell-1)}(\mathcal{G}_i, \mathcal{G}_j)$ to $\text{GNTK}^{(\ell)}(\mathcal{G}_i, \mathcal{G}_j)$) is similar to that for NTKs of MLP (Jacot et al., 2018). The final output of GNTK (without jumping knowledge) is calculated by

$$\text{GNTK}(\mathcal{G}_i, \mathcal{G}_j) = \sum_{u \in \mathcal{V}_i, u' \in \mathcal{V}_j} \left[\text{GNTK}^{(L-1)}(\mathcal{G}_i, \mathcal{G}_j)\right]_{uu'}. \tag{19}$$

### C.2  GNTK FOR NODE-LEVEL REGRESSION

We next extend the above definition of GNTK to the node-level regression setting, where the model $f(\boldsymbol{x}; \boldsymbol{\theta}, \mathcal{G})$ outputs prediction of a node, and the kernel function is defined over a pair of nodes in a single graph:

$$\text{GNTK}(\boldsymbol{x}_i, \boldsymbol{x}_j) = \phi_{gnn}(\boldsymbol{x}_i)^\top \phi_{gnn}(\boldsymbol{x}_j) = \langle \nabla_{\boldsymbol{\theta}} f(\boldsymbol{x}_i; \boldsymbol{\theta}, \mathcal{G}), \nabla_{\boldsymbol{\theta}} f(\boldsymbol{x}_j; \boldsymbol{\theta}, \mathcal{G}) \rangle, \tag{20}$$

$$\mathcal{L}(\boldsymbol{\theta}) = \frac{1}{2} \sum_{i=1}^n (y_i - f(\boldsymbol{x}_i; \boldsymbol{\theta}, \mathcal{G}))^2. \tag{21}$$

Then, the explicit formula for GNTK in node-level regression is as follows.

$$\text{GNTK}^{(0)}(\boldsymbol{x}_i, \boldsymbol{x}_j) = \boldsymbol{\Sigma}^{(0)}(\boldsymbol{x}_i, \boldsymbol{x}_j) = \boldsymbol{x}_i^\top \boldsymbol{x}_j, \tag{22}$$

Without loss of generality, we consider using random walk matrix as implementation of message passing. Then, the **message passing operation** in each layer corresponds to:

$$\mathbf{\Sigma}_{mp}^{(\ell)}\left(\boldsymbol{x}_i, \boldsymbol{x}_j\right) = \frac{1}{(|\mathcal{N}_i|+1)(|\mathcal{N}_j|+1)} \sum_{i' \in \mathcal{N}_i \cup \{i\}} \sum_{j' \in \mathcal{N}_j \cup \{j\}} \mathbf{\Sigma}^{(\ell)}\left(\boldsymbol{x}_{i'}, \boldsymbol{x}_{j'}\right)$$

$$\text{GNTK}_{mp}^{(\ell)}\left(\boldsymbol{x}_i, \boldsymbol{x}_j\right) = \frac{1}{(|\mathcal{N}_i|+1)(|\mathcal{N}_j|+1)} \sum_{i' \in \mathcal{N}_i \cup \{i\}} \sum_{j' \in \mathcal{N}_j \cup \{j\}} \text{GNTK}^{(\ell)}\left(\boldsymbol{x}_{i'}, \boldsymbol{x}_{j'}\right),$$

(23)

Moreover, the **feed-forward operation** in each layer corresponds to:

$$\mathbf{\Sigma}^{(\ell)}\left(\boldsymbol{x}_i, \boldsymbol{x}_j\right) = c \cdot \mathop{\mathbb{E}}_{u,v \sim \mathcal{N}\left(\mathbf{0}, \mathbf{\Lambda}^{(\ell)}\right)} [\sigma(u)\sigma(v)],$$

$$\dot{\mathbf{\Sigma}}^{(\ell)}\left(\boldsymbol{x}_i, \boldsymbol{x}_j\right) = c \cdot \mathop{\mathbb{E}}_{u,v \sim \mathcal{N}\left(\mathbf{0}, \mathbf{\Lambda}^{(\ell)}\right)} [\dot{\sigma}(u)\dot{\sigma}(v)],$$

$$\text{GNTK}^{(\ell)}(\boldsymbol{x}_i, \boldsymbol{x}_j) = \text{GNTK}_{mp}^{(\ell-1)}(\boldsymbol{x}_i, \boldsymbol{x}_j) \cdot \dot{\mathbf{\Sigma}}^{(\ell)}\left(\boldsymbol{x}_i, \boldsymbol{x}_j\right) + \mathbf{\Sigma}^{(\ell)}\left(\boldsymbol{x}_i, \boldsymbol{x}_j\right),$$

(24)

$$\text{where} \qquad \mathbf{\Lambda}^{(\ell)} = \left[ \begin{array}{cc} \mathbf{\Sigma}_{mp}^{(\ell-1)}(\boldsymbol{x}_i, \boldsymbol{x}_i) & \mathbf{\Sigma}_{mp}^{(\ell-1)}(\boldsymbol{x}_i, \boldsymbol{x}_j) \\ \mathbf{\Sigma}_{mp}^{(\ell-1)}(\boldsymbol{x}_j, \boldsymbol{x}_i) & \mathbf{\Sigma}_{mp}^{(\ell-1)}(\boldsymbol{x}_j, \boldsymbol{x}_j) \end{array} \right]$$

Suppose the GNN has $L$ layers and the last layer uses linear transformation that is akin to MLP, the final GNTK in node-level regression is defined as

$$\text{GNTK}(\boldsymbol{x}_i, \boldsymbol{x}_j) = \text{GNTK}_{mp}^{(L-1)}(\boldsymbol{x}_i, \boldsymbol{x}_j) \tag{25}$$

### C.3 GNTK AND FEATURE MAP FOR A TWO-LAYER GNN

We next derive the explicit NTK formula for a two-layer graph neural network in node-level regression setting. For notational convenience, we use $\boldsymbol{a}_i \in \mathbb{R}^n$ to denote adjacency, i.e.,

$$(\boldsymbol{a}_i)_j = \begin{cases} 1/(|\mathcal{N}_i|+1) & \text{if } (i,j) \in \mathcal{E} \\ 0 & \text{if } (i,j) \notin \mathcal{E} \end{cases}, \tag{26}$$

and $\mathbf{G} = \mathbf{X}\mathbf{X}^\top \in \mathbb{R}^{n \times n}$ to denote the Gram matrix of all nodes. Then we have

*(First message passing layer)*

$$\text{GNTK}^{(0)}(\boldsymbol{x}_i, \boldsymbol{x}_j) = \mathbf{\Sigma}^{(0)}\left(\boldsymbol{x}_i, \boldsymbol{x}_j\right) = \boldsymbol{x}_i^\top \boldsymbol{x}_j,$$

$$\text{GNTK}_{mp}^{(0)}(\boldsymbol{x}_i, \boldsymbol{x}_j) = \mathbf{\Sigma}_{mp}^{(0)}\left(\boldsymbol{x}_i, \boldsymbol{x}_j\right) = \boldsymbol{a}_i^\top \mathbf{G} \boldsymbol{a}_j,$$

(27)

*(First feed-forward layer)*

$$\mathbf{\Sigma}^{(1)}\left(\boldsymbol{x}_i, \boldsymbol{x}_j\right) = c \cdot \mathop{\mathbb{E}}_{u,v \sim \mathcal{N}\left(\mathbf{0}, \mathbf{\Lambda}^{(1)}\right)} [\sigma(u)\sigma(v)],$$

$$\dot{\mathbf{\Sigma}}^{(1)}\left(\boldsymbol{x}_i, \boldsymbol{x}_j\right) = c \cdot \mathop{\mathbb{E}}_{u,v \sim \mathcal{N}\left(\mathbf{0}, \mathbf{\Lambda}^{(1)}\right)} [\dot{\sigma}(u)\dot{\sigma}(v)],$$

(28)

$$\mathbf{\Lambda}^{(1)} = \left[ \begin{array}{cc} \boldsymbol{a}_i^\top \mathbf{G} \boldsymbol{a}_i & \boldsymbol{a}_i^\top \mathbf{G} \boldsymbol{a}_j \\ \boldsymbol{a}_j^\top \mathbf{G} \boldsymbol{a}_i & \boldsymbol{a}_j^\top \mathbf{G} \boldsymbol{a}_j \end{array} \right] = \left[ \begin{array}{c} \mathbf{X}^\top \boldsymbol{a}_i \\ \mathbf{X}^\top \boldsymbol{a}_j \end{array} \right] \cdot \left[ \begin{array}{cc} \mathbf{X}^\top \boldsymbol{a}_i & \mathbf{X}^\top \boldsymbol{a}_j \end{array} \right]$$

By noting that $\boldsymbol{a}_i^\top \mathbf{G} \boldsymbol{a}_j = (\mathbf{X}^\top \boldsymbol{a}_i)^\top \mathbf{X}^\top \boldsymbol{a}_j$ and substituting $\sigma(k) = k \cdot \mathbb{I}_+(k)$, $\dot{\sigma}(k) = \mathbb{I}_+(k)$, where $\mathbb{I}_+(k)$ is an indicator function that outputs 1 if $k$ is positive otherwise 0, we have the following equivalent form for the covariance

$$\mathbf{\Sigma}^{(1)}\left(\boldsymbol{x}_i, \boldsymbol{x}_j\right) = c \cdot \mathop{\mathbb{E}}_{\boldsymbol{w} \sim \mathcal{N}(\mathbf{0}, I_d)} \left[ \boldsymbol{w}^\top \mathbf{X}^\top \boldsymbol{a}_i \cdot \mathbb{I}_+(\boldsymbol{w}^\top \mathbf{X}^\top \boldsymbol{a}_i) \cdot \boldsymbol{w}^\top \mathbf{X}^\top \boldsymbol{a}_j \cdot \mathbb{I}_+(\boldsymbol{w}^\top \mathbf{X}^\top \boldsymbol{a}_j) \right],$$

$$\dot{\mathbf{\Sigma}}^{(1)}\left(\boldsymbol{x}_i, \boldsymbol{x}_j\right) = c \cdot \mathop{\mathbb{E}}_{\boldsymbol{w} \sim \mathcal{N}(\mathbf{0}, I_d)} \left[ \mathbb{I}_+(\boldsymbol{w}^\top \mathbf{X}^\top \boldsymbol{a}_i) \cdot \mathbb{I}_+(\boldsymbol{w}^\top \mathbf{X}^\top \boldsymbol{a}_j) \right].$$

(29)

Hence, we have

$$\text{GNTK}^{(1)}(\boldsymbol{x}_i, \boldsymbol{x}_j) = \text{GNTK}_{mp}^{(0)}(\boldsymbol{x}_i, \boldsymbol{x}_j) \cdot \dot{\mathbf{\Sigma}}^{(1)}\left(\boldsymbol{x}_i, \boldsymbol{x}_j\right) + \mathbf{\Sigma}^{(1)}\left(\boldsymbol{x}_i, \boldsymbol{x}_j\right)$$

$$= c \cdot \mathop{\mathbb{E}}_{\boldsymbol{w} \sim \mathcal{N}(\mathbf{0}, I_d)} \left[ \boldsymbol{a}_i^\top \mathbf{G} \boldsymbol{a}_j \cdot \mathbb{I}_+(\boldsymbol{w}^\top \mathbf{X}^\top \boldsymbol{a}_i) \cdot \mathbb{I}_+(\boldsymbol{w}^\top \mathbf{X}^\top \boldsymbol{a}_j) \right]$$

$$+ c \cdot \mathop{\mathbb{E}}_{\boldsymbol{w} \sim \mathcal{N}(\mathbf{0}, I_d)} \left[ \boldsymbol{w}^\top \mathbf{X}^\top \boldsymbol{a}_i \cdot \mathbb{I}_+(\boldsymbol{w}^\top \mathbf{X}^\top \boldsymbol{a}_i) \cdot \boldsymbol{w}^\top \mathbf{X}^\top \boldsymbol{a}_j \cdot \mathbb{I}_+(\boldsymbol{w}^\top \mathbf{X}^\top \boldsymbol{a}_j) \right]$$

(30)

*(Second message passing layer / The last layer)*

Since the GNN uses a linear transformation on top of the (second) message passing layer for output, the neural tangent kernel for a two-layer GNN is given by

$$
\begin{aligned}
\mathrm{GNTK}(\boldsymbol{x}_i, \boldsymbol{x}_j) &= \mathrm{GNTK}^{(1)}_{mp}(\boldsymbol{x}_i, \boldsymbol{x}_j) \\
&= \frac{1}{(|\mathcal{N}_i|+1)(|\mathcal{N}_j|+1)} \sum_{i' \in \mathcal{N}_i \cup \{i\}} \sum_{j' \in \mathcal{N}_j \cup \{j\}} \mathrm{GNTK}^{(1)}(\boldsymbol{x}_{i'}, \boldsymbol{x}_{j'}) \\
&= \left\langle \left[\phi^{(1)}(\boldsymbol{x}_1), \cdots, \phi^{(1)}(\boldsymbol{x}_n)\right]^{\top} \boldsymbol{a}_i, \left[\phi^{(1)}(\boldsymbol{x}_1), \cdots, \phi^{(1)}(\boldsymbol{x}_n)\right]^{\top} \boldsymbol{a}_j \right\rangle_{\mathcal{H}}
\end{aligned}
\tag{31}
$$

where $\mathbf{K}^{(1)} \in \mathbb{R}^{n \times n}$ and $\phi^{(1)} : \mathbb{R}^d \to \mathcal{H}$ is the kernel matrix and feature map induced by $\mathrm{GNTK}^{(1)}$. By Eq. 31, the final feature map $\phi_{gnn}(\boldsymbol{x})$ is

$$
\phi_{gnn}(\boldsymbol{x}_i) = \left[\phi^{(1)}(\boldsymbol{x}_1), \cdots, \phi^{(1)}(\boldsymbol{x}_n)\right]^{\top} \boldsymbol{a}_i.
\tag{32}
$$

Also, notice that in Eq. 30,

$$
\begin{aligned}
\boldsymbol{a}_i^{\top} \mathbf{G} \boldsymbol{a}_j \cdot \mathbb{I}_+(\boldsymbol{w}^{\top}\mathbf{X}^{\top}\boldsymbol{a}_i) \cdot \mathbb{I}_+(\boldsymbol{w}^{\top}\mathbf{X}^{\top}\boldsymbol{a}_j) &= \phi_i^{\top}\phi_j, \\
\boldsymbol{w}^{\top}\mathbf{X}^{\top}\boldsymbol{a}_i \cdot \mathbb{I}_+(\boldsymbol{w}^{\top}\mathbf{X}^{\top}\boldsymbol{a}_i) \cdot \boldsymbol{w}^{\top}\mathbf{X}^{\top}\boldsymbol{a}_j \cdot \mathbb{I}_+(\boldsymbol{w}^{\top}\mathbf{X}^{\top}\boldsymbol{a}_j) &= (\boldsymbol{w}^{\top}\phi_i)^{\top}\boldsymbol{w}^{\top}\phi_j, \\
\text{where} \quad \phi_i = \mathbf{X}^{\top}\boldsymbol{a}_i \cdot \mathbb{I}_+(\boldsymbol{w}^{\top}\mathbf{X}^{\top}\boldsymbol{a}_i). &
\end{aligned}
\tag{33}
$$

Then, the feature map $\phi^{(1)}$ can be written as

$$
\begin{aligned}
\phi^{(1)}(\boldsymbol{x}_i) = c' \cdot \Big[ &\mathbf{X}^{\top}\boldsymbol{a}_i \cdot \mathbb{I}_+\left(\boldsymbol{w}^{(1)\top}\mathbf{X}^{\top}\boldsymbol{a}_i\right), \boldsymbol{w}^{(1)\top}\mathbf{X}^{\top}\boldsymbol{a}_i \cdot \mathbb{I}_+\left(\boldsymbol{w}^{(1)\top}\mathbf{X}^{\top}\boldsymbol{a}_i\right), \\
&\mathbf{X}^{\top}\boldsymbol{a}_i \cdot \mathbb{I}_+\left(\boldsymbol{w}^{(2)\top}\mathbf{X}^{\top}\boldsymbol{a}_i\right), \boldsymbol{w}^{(2)\top}\mathbf{X}^{\top}\boldsymbol{a}_i \cdot \mathbb{I}_+\left(\boldsymbol{w}^{(2)\top}\mathbf{X}^{\top}\boldsymbol{a}_i\right), \\
&\cdots \\
&\mathbf{X}^{\top}\boldsymbol{a}_i \cdot \mathbb{I}_+\left(\boldsymbol{w}^{(\infty)\top}\mathbf{X}^{\top}\boldsymbol{a}_i\right), \boldsymbol{w}^{(\infty)\top}\mathbf{X}^{\top}\boldsymbol{a}_i \cdot \mathbb{I}_+\left(\boldsymbol{w}^{(\infty)\top}\mathbf{X}^{\top}\boldsymbol{a}_i\right) \Big]
\end{aligned}
\tag{34}
$$

where $\boldsymbol{w}^{(k)} \sim \mathcal{N}(\mathbf{0}, I_d)$ is random Gaussian vector in $\mathbb{R}^d$, with the superscript $^{(k)}$ denoting that it is the $k$-th sample among infinitely many i.i.d. sampled ones, $c'$ is a constant. We write Eq. 34 in short as

$$
\phi^{(1)}(\boldsymbol{x}_i) = c' \cdot \left[\mathbf{X}^{\top}\boldsymbol{a}_i \cdot \mathbb{I}_+\left(\boldsymbol{w}^{(k)\top}\mathbf{X}^{\top}\boldsymbol{a}_i\right), \boldsymbol{w}^{(k)\top}\mathbf{X}^{\top}\boldsymbol{a}_i \cdot \mathbb{I}_+\left(\boldsymbol{w}^{(k)\top}\mathbf{X}^{\top}\boldsymbol{a}_i\right), \cdots\right].
\tag{35}
$$

Finally, substituting Eq. 35 into Eq. 32 completes the proof.

## D  PROOF FOR THEOREM 4 AND THEOREM 5

To analyse the extrapolation behavior of PMLP along a certain direction $\boldsymbol{v}$ in the testing phase and compare it to MLP, we consider a newly arrived testing node $\boldsymbol{x}_0 = t\boldsymbol{v}$, whose degree (with self-connection) is $\tilde{d}$ and its corresponding adjacency vector is $\boldsymbol{a} \in \mathbb{R}^{n+1}$, where $(\boldsymbol{a})_i = 1/\tilde{d}$ if $(i, 0) \in \mathcal{E}$ otherwise 0. Following (Xu et al., 2021; Bietti & Mairal, 2019), we consider a constant bias term and denote the data $\boldsymbol{x}$ plus this term as $\hat{\boldsymbol{x}} = [\boldsymbol{x}|1]$. Then, the asymptotic behavior of $f(\cdot)$ at large distances from the training data range can be characterized by the change of network output with a fixed-length step $\Delta t \cdot \boldsymbol{v}$ along the direction $\boldsymbol{v}$, which is given by the following in the NTK regime

$$
\frac{1}{\Delta t}\left(f_{mlp}(\hat{\boldsymbol{x}}) - f_{mlp}(\hat{\boldsymbol{x}}_0)\right) = \frac{1}{\Delta t}\boldsymbol{w}^{*\top}_{mlp}(\phi_{mlp}(\hat{\boldsymbol{x}}) - \phi_{mlp}(\hat{\boldsymbol{x}}_0))
\tag{36}
$$

$$
\frac{1}{\Delta t}\left(f_{pmlp}(\hat{\boldsymbol{x}}) - f_{pmlp}(\hat{\boldsymbol{x}}_0)\right) = \frac{1}{\Delta t}\boldsymbol{w}^{*\top}_{mlp}(\phi_{gnn}(\hat{\boldsymbol{x}}) - \phi_{gnn}(\hat{\boldsymbol{x}}_0))
\tag{37}
$$

where $\boldsymbol{x} = \boldsymbol{x}_0 + \Delta t \cdot \boldsymbol{v} = (t + \Delta t)\boldsymbol{v}$, $\hat{\boldsymbol{x}} = [\hat{\boldsymbol{x}}|1]$ and $\hat{\boldsymbol{x}}_0 = [\hat{\boldsymbol{x}}_0|1]$. As our interest is in how PMLP extrapolation, we use $f(\cdot)$ to refer to $f_{pmlp}(\cdot)$ in the rest of the proof. By Lemma. 2, the explicit formula for computing this node's feature map is given by

$$
\phi_{gnn}(\hat{\boldsymbol{x}}_0) = \left[\phi^{(1)}(\hat{\boldsymbol{x}}_0), \phi^{(1)}(\boldsymbol{x}_1; \hat{\boldsymbol{x}}_0), \cdots, \phi^{(1)}(\boldsymbol{x}_n; \hat{\boldsymbol{x}}_0)\right]^{\top} \boldsymbol{a},
\tag{38}
$$

where

$$\phi^{(1)}(\hat{\boldsymbol{x}}_0) = c' \cdot \left[ [\hat{\boldsymbol{x}}_0, \mathbf{X}]^\top \boldsymbol{a} \cdot \mathbb{I}_+ \left( \boldsymbol{w}^{(k)^\top} [\hat{\boldsymbol{x}}_0, \mathbf{X}]^\top \boldsymbol{a} \right), \right.$$
$$\left. \boldsymbol{w}^{(k)^\top} [\hat{\boldsymbol{x}}_0, \mathbf{X}]^\top \boldsymbol{a} \cdot \mathbb{I}_+ \left( \boldsymbol{w}^{(k)^\top} [\hat{\boldsymbol{x}}_0, \mathbf{X}]^\top \boldsymbol{a} \right), \dots \right], \tag{39}$$

and similarly

$$\phi^{(1)}(\boldsymbol{x}_i; \hat{\boldsymbol{x}}_0) = c' \cdot \left[ [\hat{\boldsymbol{x}}_0, \mathbf{X}]^\top \boldsymbol{a}_i \cdot \mathbb{I}_+ \left( \boldsymbol{w}^{(k)^\top} [\hat{\boldsymbol{x}}_0, \mathbf{X}]^\top \boldsymbol{a}_i \right), \right.$$
$$\left. \boldsymbol{w}^{(k)^\top} [\hat{\boldsymbol{x}}_0, \mathbf{X}]^\top \boldsymbol{a}_i \cdot \mathbb{I}_+ \left( \boldsymbol{w}^{(k)^\top} [\hat{\boldsymbol{x}}_0, \mathbf{X}]^\top \boldsymbol{a}_i \right), \dots \right], \tag{40}$$

$\boldsymbol{w}^{(k)} \sim \mathcal{N}(\mathbf{0}, \boldsymbol{I}_d)$, with $k$ going to infinity, $c'$ is a constant, $\mathbb{I}_+(k)$ is an indicator function that outputs 1 if $k$ is positive otherwise 0, and $(\boldsymbol{a}_i)_1 = 1/(|\mathcal{N}_i| + 1)$ if $i$ is connected to the new testing node. It follows from Eq. 37 that

$$\frac{1}{\Delta t} \left( f(\hat{\boldsymbol{x}}) - f(\hat{\boldsymbol{x}}_0) \right)$$
$$= \frac{1}{\Delta t} \boldsymbol{w}_{mlp}^{*\top} \left[ \phi^{(1)}(\hat{\boldsymbol{x}}) - \phi^{(1)}(\hat{\boldsymbol{x}}_0), \cdots, \phi^{(1)}(\boldsymbol{x}_n; \hat{\boldsymbol{x}}) - \phi^{(1)}(\boldsymbol{x}_n; \hat{\boldsymbol{x}}_0) \right]^\top \boldsymbol{a} \tag{41}$$
$$= \frac{1}{\tilde{d}\Delta t} \boldsymbol{w}_{mlp}^{*\top} \left( \phi^{(1)}(\hat{\boldsymbol{x}}) - \phi^{(1)}(\hat{\boldsymbol{x}}_0) \right) + \frac{1}{\tilde{d}\Delta t} \sum_{i \in \mathcal{N}_0} \boldsymbol{w}_{mlp}^{*\top} \left( \phi^{(1)}(\boldsymbol{x}_i; \hat{\boldsymbol{x}}) - \phi^{(1)}(\boldsymbol{x}_i; \hat{\boldsymbol{x}}_0) \right)$$

Now, let us consider $\boldsymbol{w}_{mlp}^{*\top} \left( \phi^{(1)}(\hat{\boldsymbol{x}}) - \phi^{(1)}(\hat{\boldsymbol{x}}_0) \right)$. Recall that $\boldsymbol{w}_{mlp}^{*\top}$ is from infinite dimensional Hilbert space, and $\boldsymbol{w}^{(k)}$ is drawn from Gaussian with $k$ going to infinity in Eq. 39 and Eq. 40, where each $\boldsymbol{w}^{(k)}$ corresponds to some certain dimensions of $\boldsymbol{w}_{mlp}^*$. Let us denote the part that corresponds to the first line in Eq. 39 as $\boldsymbol{\beta}_{\boldsymbol{w}^{(k)}}$ and the second line in Eq. 39 as $\boldsymbol{\gamma}_{\boldsymbol{w}^{(k)}}$. Consider the following way of rearrangement for (the first element in) $\phi^{(1)}(\hat{\boldsymbol{x}}) - \phi^{(1)}(\hat{\boldsymbol{x}}_0)$

$$[\hat{\boldsymbol{x}}, \mathbf{X}]^\top \boldsymbol{a} \cdot \mathbb{I}_+ \left( \boldsymbol{w}^{(k)^\top} [\hat{\boldsymbol{x}}, \mathbf{X}]^\top \boldsymbol{a} \right) - [\hat{\boldsymbol{x}}_0, \mathbf{X}]^\top \boldsymbol{a} \cdot \mathbb{I}_+ \left( \boldsymbol{w}^{(k)^\top} [\hat{\boldsymbol{x}}_0, \mathbf{X}]^\top \boldsymbol{a} \right)$$
$$= [\hat{\boldsymbol{x}}, \mathbf{X}]^\top \boldsymbol{a} \left( \mathbb{I}_+ \left( \boldsymbol{w}^{(k)^\top} [\hat{\boldsymbol{x}}, \mathbf{X}]^\top \boldsymbol{a} \right) - \mathbb{I}_+ \left( \boldsymbol{w}^{(k)^\top} [\hat{\boldsymbol{x}}_0, \mathbf{X}]^\top \boldsymbol{a} \right) \right) \tag{42}$$
$$+ \frac{1}{\tilde{d}} [\Delta t \boldsymbol{v} \mid 0] \cdot \mathbb{I}_+ \left( \boldsymbol{w}^{(k)^\top} [\hat{\boldsymbol{x}}_0, \mathbf{X}]^\top \boldsymbol{a} \right),$$

where $\frac{1}{\tilde{d}} [\Delta t \boldsymbol{v} \mid 0]$ is obtained by subtracting $[\hat{\boldsymbol{x}}_0, \mathbf{X}]^\top \boldsymbol{a}$ from $[\hat{\boldsymbol{x}}, \mathbf{X}]^\top \boldsymbol{a}$. Then, we can re-write $\boldsymbol{w}_{mlp}^{*\top} \left( \phi^{(1)}(\hat{\boldsymbol{x}}) - \phi^{(1)}(\hat{\boldsymbol{x}}_0) \right)$ into a more convenient form:

$$\frac{1}{\Delta t} \boldsymbol{w}_{mlp}^{*\top} \left( \phi^{(1)}(\hat{\boldsymbol{x}}) - \phi^{(1)}(\hat{\boldsymbol{x}}_0) \right) \tag{43}$$

$$= \int \frac{1}{\tilde{d}} \boldsymbol{\beta}_{\boldsymbol{w}}^\top [\boldsymbol{v} \mid 0] \cdot \mathbb{I}_+ \left( \boldsymbol{w}^\top [\hat{\boldsymbol{x}}_0, \mathbf{X}]^\top \boldsymbol{a} \right) \mathrm{d}\mathbb{P}(\boldsymbol{w}) \tag{44}$$

$$+ \int \boldsymbol{\beta}_{\boldsymbol{w}}^\top [\hat{\boldsymbol{x}}/\Delta t, \mathbf{X}/\Delta t]^\top \boldsymbol{a} \left( \mathbb{I}_+ \left( \boldsymbol{w}^\top [\hat{\boldsymbol{x}}, \mathbf{X}]^\top \boldsymbol{a} \right) - \mathbb{I}_+ \left( \boldsymbol{w}^\top [\hat{\boldsymbol{x}}_0, \mathbf{X}]^\top \boldsymbol{a} \right) \right) \mathrm{d}\mathbb{P}(\boldsymbol{w}) \tag{45}$$

$$+ \int \frac{1}{\tilde{d}} \boldsymbol{\gamma}_{\boldsymbol{w}} \boldsymbol{w}^\top [\boldsymbol{v} \mid 0] \cdot \mathbb{I}_+ \left( \boldsymbol{w}^\top [\hat{\boldsymbol{x}}_0, \mathbf{X}]^\top \boldsymbol{a} \right) \mathrm{d}\mathbb{P}(\boldsymbol{w}) \tag{46}$$

$$+ \int \boldsymbol{\gamma}_{\boldsymbol{w}} \boldsymbol{w}^\top [\hat{\boldsymbol{x}}/\Delta t, \mathbf{X}/\Delta t]^\top \boldsymbol{a} \left( \mathbb{I}_+ \left( \boldsymbol{w}^\top [\hat{\boldsymbol{x}}, \mathbf{X}]^\top \boldsymbol{a} \right) - \mathbb{I}_+ \left( \boldsymbol{w}^\top [\hat{\boldsymbol{x}}_0, \mathbf{X}]^\top \boldsymbol{a} \right) \right) \mathrm{d}\mathbb{P}(\boldsymbol{w}) \tag{47}$$

**Remark.** For other components in Eq. 41, i.e., $\frac{1}{\Delta t} \boldsymbol{w}_{mlp}^{*\top} \left( \phi^{(1)}(\boldsymbol{x}_i; \hat{\boldsymbol{x}}) - \phi^{(1)}(\boldsymbol{x}_i; \hat{\boldsymbol{x}}_0) \right)$ where $i \in \mathcal{N}_0$, the corresponding result of the expansion in Eq. 43 is similar, which only differs by a scaling factor (since the first element in both $\boldsymbol{a}$ and $\boldsymbol{a}_i$ indicating whether the current node is connected to the new testing node is non-zero) and replacing $\boldsymbol{a}$ with $\boldsymbol{a}_i$. Therefore, in the following proof we focus on Eq. 43 and then generalize the result to other components in Eq. 41.

### D.1 PROOF FOR THEOREM 4 (CONVERGENCE TO A LINEAR FUNCTION)

We first analyse the convergence of Eq. 43. Specifically, for Eq. 44:

$$\int \frac{1}{\tilde{d}} \boldsymbol{\beta}_{\boldsymbol{w}}^\top [\boldsymbol{v} \mid 0] \cdot \mathbb{I}_+ \left( \boldsymbol{w}^\top [\hat{\boldsymbol{x}}_0, \mathbf{X}]^\top \boldsymbol{a} \right) d\mathbb{P}(\boldsymbol{w})$$

$$= \int \frac{1}{\tilde{d}} \boldsymbol{\beta}_{\boldsymbol{w}}^\top [\boldsymbol{v} \mid 0] \cdot \mathbb{I}_+ \left( \boldsymbol{w}^\top [\hat{\boldsymbol{x}}_0/t, \mathbf{X}/t]^\top \boldsymbol{a} \right) d\mathbb{P}(\boldsymbol{w}) \tag{48}$$

$$\to \int \frac{1}{\tilde{d}} \boldsymbol{\beta}_{\boldsymbol{w}}^\top [\boldsymbol{v} \mid 0] \cdot \mathbb{I}_+ \left( \boldsymbol{w}^\top [\boldsymbol{v} \mid 0] \right) d\mathbb{P}(\boldsymbol{w}) = \frac{c'_{\boldsymbol{v}}}{\tilde{d}}, \quad \text{as } t \to \infty$$

where the final result is a constant that depends on training data, direction $\boldsymbol{v}$ and node degree $\tilde{d}$. Moreover, the convergence of Eq. 45 is given by

$$\int \boldsymbol{\beta}_{\boldsymbol{w}}^\top [\hat{\boldsymbol{x}}/\Delta t, \mathbf{X}/\Delta t]^\top \boldsymbol{a} \left( \mathbb{I}_+ \left( \boldsymbol{w}^\top [\hat{\boldsymbol{x}}, \mathbf{X}]^\top \boldsymbol{a} \right) - \mathbb{I}_+ \left( \boldsymbol{w}^\top [\hat{\boldsymbol{x}}_0, \mathbf{X}]^\top \boldsymbol{a} \right) \right) d\mathbb{P}(\boldsymbol{w})$$

$$= \int \boldsymbol{\beta}_{\boldsymbol{w}}^\top [\hat{\boldsymbol{x}}/\Delta t, \mathbf{X}/\Delta t]^\top \boldsymbol{a} \left( \mathbb{I}_+ \left( \boldsymbol{w}^\top [[\boldsymbol{v} \mid \frac{1}{t + \Delta t}], \frac{\mathbf{X}}{t + \Delta t}]^\top \boldsymbol{a} \right) - \mathbb{I}_+ \left( \boldsymbol{w}^\top [[\boldsymbol{v} \mid \frac{1}{t}], \frac{\mathbf{X}}{t}]^\top \boldsymbol{a} \right) \right) d\mathbb{P}(\boldsymbol{w})$$

$$\to \int \boldsymbol{\beta}_{\boldsymbol{w}}^\top [\hat{\boldsymbol{x}}/\Delta t, \mathbf{X}/\Delta t]^\top \boldsymbol{a} \left( \mathbb{I}_+ \left( \boldsymbol{w}^\top [\boldsymbol{v} \mid 0] \right) - \mathbb{I}_+ \left( \boldsymbol{w}^\top [\boldsymbol{v} \mid 0] \right) \right) d\mathbb{P}(\boldsymbol{w}) = 0, \quad \text{as } t \to \infty$$

$$\tag{49}$$

The similar results in Eq. 48 and Eq. 49 also apply to analysis of Eq. 46 and Eq. 47, respectively. By combining these results, we conclude that

$$\frac{1}{\Delta t} \boldsymbol{w}_{mlp}^{*\top} \left( \phi^{(1)}(\hat{\boldsymbol{x}}) - \phi^{(1)}(\hat{\boldsymbol{x}}_0) \right) \to \frac{c_{\boldsymbol{v}}}{\tilde{d}}, \quad \text{as } t \to \infty \tag{50}$$

It follows that

$$\frac{1}{\Delta t} \left( f(\hat{\boldsymbol{x}}) - f(\hat{\boldsymbol{x}}_0) \right) \to c_{\boldsymbol{v}} \tilde{d}^{-1} \sum_{i \in \mathcal{N}_0 \cup \{0\}} \tilde{d}_i^{-1}, \quad \text{as } t \to \infty. \tag{51}$$

In conclusion, both MLP and PMLP with ReLU activation will eventually converge to a linear function along directions away from the training data. In fact, this result also holds true for two-layer GNNs with weighted-sum style message passing layers by simply replacing $\boldsymbol{w}_{mlp}^*$ by $\boldsymbol{w}_{gnn}^*$ in the proof.

However, a remarkable difference between MLP and PMLP is that the linear coefficient for MLP is a constant $c_{\boldsymbol{v}}$ that is fixed upon a specific direction $\boldsymbol{v}$ and not affected by the inter-connection between testing node $\boldsymbol{x}$ and training data $\{(\boldsymbol{x}_i, y_i)\}_{i=1}^n$. In contrast, the linear coefficient for PMLP (and GNN) is also dependent on testing node's degree and the degrees of adjacent nodes.

Moreover, by Proposition. 1, MLP and PMLP share the same $\boldsymbol{w}_{mlp}^*$ (including $\boldsymbol{\beta}_{\boldsymbol{w}}$ and $\boldsymbol{\gamma}_{\boldsymbol{w}}$), and thus the constant $c_{\boldsymbol{v}}$ in Eq. 51 is exactly the linear coefficient of MLP. This can also be verified by setting $\boldsymbol{x}$ to be an isolated node, in which case $\tilde{d}^{-1} \sum_{i \in \mathcal{N}_0 \cup \{0\}} \tilde{d}_i^{-1} = 1$ and PMLP is equivalent to MLP. Therefore, we can directly compare the linear coefficients for MLP and PMLP. As an immediate consequence, if all node degrees of adjacent nodes are larger than the node degree of the testing node, the linear coefficient will become smaller, vice versa.

### D.2 PROOF FOR THEOREM 5

We next analyse the convergence rate for Eq. 48 and Eq. 49 to see to what extent can PMLP deviate from the converged linear coefficient as an indication of its tolerance to out-of-distribution sample. For Eq. 48, we have

$$\left| \int \frac{1}{\tilde{d}} \boldsymbol{\beta}_{\boldsymbol{w}}^\top [\boldsymbol{v} \mid 0] \cdot \left( \mathbb{I}_+ \left( \boldsymbol{w}^\top [\hat{\boldsymbol{x}}_0, \mathbf{X}]^\top \boldsymbol{a} \right) - \mathbb{I}_+ \left( \boldsymbol{w}^\top [\boldsymbol{v} \mid 0] \right) \right) d\mathbb{P}(\boldsymbol{w}) \right|$$

$$\leq \frac{c'_1}{\tilde{d}} \cdot \int \left| \mathbb{I}_+ \left( \boldsymbol{w}^\top [\hat{\boldsymbol{x}}_0, \mathbf{X}]^\top \boldsymbol{a} \right) - \mathbb{I}_+ \left( \boldsymbol{w}^\top [\boldsymbol{v} \mid 0] \right) \right| d\mathbb{P}(\boldsymbol{w}) \tag{52}$$

$$= \frac{c'_1}{\tilde{d}} \cdot \int \left| \mathbb{I}_+ \left( \boldsymbol{w}^\top [[\boldsymbol{x}_0 \mid 1], \mathbf{X}]^\top \boldsymbol{a} \right) - \mathbb{I}_+ \left( \boldsymbol{w}^\top [\boldsymbol{x}_0 \mid 0] \right) \right| d\mathbb{P}(\boldsymbol{w})$$

Based on the observation that the integral of $|\mathbb{I}_+(\boldsymbol{w}^\top \boldsymbol{v}_1) - \mathbb{I}_+(\boldsymbol{w}^\top \boldsymbol{v}_2)|$ represents the volume of non-overlapping part of two half-balls that are orthogonal to $\boldsymbol{v}_1$ and $\boldsymbol{v}_2$, which grows linearly with the angle between $\boldsymbol{v}_1$ and $\boldsymbol{v}_2$, denoted by $\angle(\boldsymbol{v}_1, \boldsymbol{v}_2)$. Therefore, we have

$$
\begin{aligned}
& \frac{c_1'}{\tilde{d}} \cdot \int \left| \mathbb{I}_+ \left( \boldsymbol{w}^\top [[\boldsymbol{x}_0 \mid 1], \mathbf{X}]^\top \boldsymbol{a} \right) - \mathbb{I}_+ \left( \boldsymbol{w}^\top [\boldsymbol{x}_0 \mid 0] \right) \right| d\mathbb{P}(\boldsymbol{w}) \\
= \; & \frac{c_1}{\tilde{d}} \cdot \angle \left( [[\boldsymbol{x}_0 \mid 1], \mathbf{X}]^\top \boldsymbol{a} \, , \; [\boldsymbol{x}_0 \mid 0] \right),
\end{aligned}
\tag{53}
$$

Note that the first term in the angle can be decomposed as

$$
\tilde{d} \cdot [[\boldsymbol{x}_0 \mid 1], \mathbf{X}]^\top \boldsymbol{a} = [\boldsymbol{x}_0 \mid 0] + [\mathbf{0} \mid 1] + \sum_{i \in \mathcal{N}(0)} \boldsymbol{x}_i.
\tag{54}
$$

Suppose all node features are normalized, then we have

$$
\begin{aligned}
\angle \left( [[\boldsymbol{x}_0, 1], \mathbf{X}]^\top \boldsymbol{a} \, , \; [\boldsymbol{x}_0, 0] \right) & \leq \angle \left( [\boldsymbol{x}_0, 0] \, , \; [\boldsymbol{x}_0, 1] \right) + \angle (\hat{\boldsymbol{x}}_0 \, , \; \hat{\boldsymbol{x}}_0 + \sum_{i \in \mathcal{N}(0)} \boldsymbol{x}_i) \\
& = \arctan(\frac{1}{t}) + \arctan(\frac{(\tilde{d} - 1)\sqrt{1 - \alpha^2}}{(\tilde{d} - 1)\alpha + \sqrt{t^2 + 1}}) \\
& = O(\frac{1 + (\tilde{d} - 1)\sqrt{1 - \alpha^2}}{t})
\end{aligned}
\tag{55}
$$

where $\alpha$ denotes the cosine similarity of the testing node and the sum of its neighbors. The last step is obtained by noting $\arctan(x) < x$. Using the same reasoning, for Eq. 49, we have

$$
\begin{aligned}
& \left| \int \boldsymbol{\beta}_{\boldsymbol{w}}^\top [\hat{\boldsymbol{x}}/\Delta t, \mathbf{X}/\Delta t]^\top \boldsymbol{a} \left( \mathbb{I}_+ \left( \boldsymbol{w}^\top [\hat{\boldsymbol{x}}_0, \mathbf{X}]^\top \boldsymbol{a} \right) - \mathbb{I}_+ \left( \boldsymbol{w}^\top [\hat{\boldsymbol{x}}, \mathbf{X}]^\top \boldsymbol{a} \right) \right) d\mathbb{P}(\boldsymbol{w}) \right| \\
\leq \; & |\boldsymbol{\beta}_{\boldsymbol{w}}^\top [\hat{\boldsymbol{x}}/\Delta t, \mathbf{X}/\Delta t]^\top \boldsymbol{a}| \cdot \int \left| \mathbb{I}_+ \left( \boldsymbol{w}^\top [\hat{\boldsymbol{x}}_0, \mathbf{X}]^\top \boldsymbol{a} \right) - \mathbb{I}_+ \left( \boldsymbol{w}^\top [\hat{\boldsymbol{x}}, \mathbf{X}]^\top \boldsymbol{a} \right) \right| d\mathbb{P}(\boldsymbol{w}) \\
= \; & |\boldsymbol{\beta}_{\boldsymbol{w}}^\top [\hat{\boldsymbol{x}}/\Delta t, \mathbf{X}/\Delta t]^\top \boldsymbol{a}| \cdot \int \left| \mathbb{I}_+ \left( \boldsymbol{w}^\top \left[ [\boldsymbol{x}_0 \mid 1], \mathbf{X} \right]^\top \boldsymbol{a} \right) - \mathbb{I}_+ \left( \boldsymbol{w}^\top \left[ [\boldsymbol{x}_0 \mid \frac{t}{t + \Delta t}], \frac{t}{t + \Delta t} \mathbf{X} \right]^\top \boldsymbol{a} \right) \right| d\mathbb{P}(\boldsymbol{w}) \\
= \; & |\boldsymbol{\beta}_{\boldsymbol{w}}^\top [\hat{\boldsymbol{x}}/\Delta t, \mathbf{X}/\Delta t]^\top \boldsymbol{a}| \cdot \angle \left( \left[ [\boldsymbol{x}_0 \mid 1], \mathbf{X} \right]^\top \boldsymbol{a} \, , \; \left[ [\boldsymbol{x}_0 \mid \frac{t}{t + \Delta t}], \frac{t}{t + \Delta t} \mathbf{X} \right]^\top \boldsymbol{a} \right),
\end{aligned}
\tag{56}
$$

Note that the second term in the angle can be re-written as

$$
\left[ [\boldsymbol{x}_0 \mid \frac{t}{t + \Delta t}], \frac{t}{t + \Delta t} \mathbf{X} \right]^\top \boldsymbol{a} = \frac{t}{t + \Delta t} \cdot \left[ [\boldsymbol{x}_0 \mid 1], \mathbf{X} \right]^\top \boldsymbol{a} + \frac{\Delta t}{t + \Delta t} \cdot \left[ [\boldsymbol{x}_0 \mid 0], \mathbf{0} \right]^\top \boldsymbol{a},
\tag{57}
$$

and hence the angle is at most $\Delta t/(t + \Delta t)$ times of that in Eq. 53. It follows that

$$
\begin{aligned}
& \left| \int \boldsymbol{\beta}_{\boldsymbol{w}}^\top [\hat{\boldsymbol{x}}/\Delta t, \mathbf{X}/\Delta t]^\top \boldsymbol{a} \left( \mathbb{I}_+ \left( \boldsymbol{w}^\top [\hat{\boldsymbol{x}}_0, \mathbf{X}]^\top \boldsymbol{a} \right) - \mathbb{I}_+ \left( \boldsymbol{w}^\top [\hat{\boldsymbol{x}}, \mathbf{X}]^\top \boldsymbol{a} \right) \right) d\mathbb{P}(\boldsymbol{w}) \right| \\
= \; & O(\frac{t + \Delta t}{\tilde{d}}) \cdot O(\frac{1 + (\tilde{d} - 1)\sqrt{1 - \alpha^2}}{t}) \cdot O(\frac{\Delta t}{t + \Delta t}) \\
= \; & O(\frac{1 + (\tilde{d} - 1)\sqrt{1 - \alpha^2}}{\tilde{d} t})
\end{aligned}
\tag{58}
$$

For Eq. 46 and Eq. 47, similar results can be derived by bounding $\boldsymbol{w}$ with standard concentration techniques. By substituting the above convergence rates to Eq. 43, and dividing it by the linear coefficient in Eq. 50, the convergence rate for Eq. 43 is

$$
\left| \frac{\frac{1}{\Delta t} \boldsymbol{w}_{mlp}^{*\top} \left( \phi^{(1)}(\hat{\boldsymbol{x}}) - \phi^{(1)}(\hat{\boldsymbol{x}}_0) \right) - c_{\boldsymbol{v}}/\tilde{d}}{c_{\boldsymbol{v}}/\tilde{d}} \right| = O(\frac{1 + (\tilde{d} - 1)\sqrt{1 - \alpha^2}}{t})
\tag{59}
$$

It follows that the convergence rate for Eq. 37 is $O(\frac{1 + (\tilde{d}_{max} - 1)\sqrt{1 - \alpha_{min}^2}}{t})$, where $\tilde{d}_{max}$ denotes the maximum node degree in the testing node' neighbors (including itself) and

$$
\alpha_{min} = \min\{\alpha_i\}_{i \in \mathcal{N}_0 \cup \{0\}}
\tag{60}
$$

denotes the minimum cosine similarity for the testing node' neighbors (including itself). This completes the proof.

# E OVER-SMOOTHING, DEEP GNNS, AND HETEROPHILY

**Over-Smoothing and Deep GNNs.** To gain more insights into the over-smoothing problem and the impact of model depth, we further investigate GNN architectures with residual connections (including GCN-style ones where residual connections are employed across FF layers, e.g., JKNet (Xu et al., 2018b) and GCNII (Chen et al., 2020b), and SGC/APPNP-style ones where the implementation of MP layers involve residual connections, e.g., APPNP with non-zero $\alpha$). Table. 3 and Table. 4 respectively report the results for SGC/APPNP-style and GCN-style GNNs on Cora dataset. Similar trends are also observed on other datasets, some of which are plotted in Fig. 5.

As we can see, the performances of PMLPs without residual connections (i.e., $\text{PMLP}_{GCN}$, $\text{PMLP}_{SGC}$ and $\text{PMLP}_{APP}$) exhibit very similar downward trends with their GNN counterparts w.r.t. increasing layer number (from 2 to 128). Such phenomenon in GNNs is commonly thought to be caused by the oversmoothing issue wherein node features become hard to distinguish after multiple steps of message passing, and since PMLP is immune from such problem in the training stage but still perform poorly in testing, we may conclude that oversmoothing is more of a problem related to failure modes of GNNs' generalization ability, rather than impairing their representational power, which is somewhat in alignment with (Cong et al., 2021) where the authors theoretically show very deep GNNs that are vulnerable to oversmoothing can still achieve high training accuracy but will perform poorly in terms of generalization. We experimental results further suggest the reason why additional residual connections (either in MP layers, e.g., ResAPPNP/SGC, or across FF layers, e.g., GCNII and JKNet) are empirical effective for solving oversmoothing is that they can improve GNN's generalization ability according to results of $\text{PMLP}_{GCNII}$, $\text{PMLP}_{JKNet}$, $\text{PMLP}_{ResSGC}$ and $\text{PMLP}_{ResAPP}$ where model depth seems to have less impact on their generalization performance.

**Graph Heterophily.** We further conduct experiments on six datasets (i.e., Chameleon, Squirrel, Film, Cornell, Texas, Wisconsin (Pei et al., 2020)) with high heterophily levels, and the results are shown in Table. 5. As we can see, PMLP achieves better performance than MLP when its GNN counterpart can outperform MLP, but is otherwise inferior than MLP. This indicates that the inherent generalization ability of a certain GNN architecture is also related to whether the message passing scheme is suitable for the characteristics of data, which also partially explains why training more dedicated GNN architectures such as H2GCN (Zhu et al., 2020) is helpful for improving model's generalization performance on heterophilic graphs.

Table 3: For SGC and APPNP styles, layer number denotes the number of MP layers. The number of FF layers and hidden size are fixed as 2 and 64. 'Res' denotes using residual connection in the form of $\boldsymbol{X}^{(k)} = (1 - \alpha)\,\mathrm{MP}(\boldsymbol{X}^{(k-1)}) + \alpha\boldsymbol{X}^{(0)}$.

| | # Layers | 2 | 4 | 8 | 16 | 32 | 64 | 128 |
|---|---|---|---|---|---|---|---|---|
| | MLP | $52.56 \pm 0.41$ | $52.56 \pm 0.41$ | $52.56 \pm 0.41$ | $52.56 \pm 0.41$ | $52.56 \pm 0.41$ | $52.56 \pm 0.41$ | $52.56 \pm 0.41$ |
| $\alpha = 0$ | SGC | $73.12 \pm 1.20$ | $75.36 \pm 1.40$ | $76.78 \pm 1.63$ | $75.22 \pm 2.08$ | $73.08 \pm 2.75$ | $64.70 \pm 4.14$ | $47.64 \pm 3.09$ |
| | $\mathrm{PMLP}_{SGC}$ | $73.48 \pm 1.30$ | $75.38 \pm 1.73$ | $76.20 \pm 2.12$ | $75.66 \pm 2.18$ | $73.60 \pm 3.07$ | $67.76 \pm 4.38$ | $53.64 \pm 6.35$ |
| $\alpha = 0.1$ | SGC+Res | $71.56 \pm 1.11$ | $73.98 \pm 1.16$ | $75.02 \pm 1.28$ | $75.50 \pm 1.05$ | $75.40 \pm 1.05$ | $75.38 \pm 1.05$ (Converged) | $75.38 \pm 1.05$ (Converged) |
| | $\mathrm{PMLP}_{SGC+Res}$ | $72.80 \pm 0.67$ | $75.08 \pm 1.14$ | $76.06 \pm 1.13$ | $76.22 \pm 1.18$ | $76.20 \pm 1.23$ | $76.14 \pm 1.14$ | $76.14 \pm 1.14$ (Converged) |
| | MLP | $52.56 \pm 0.41$ | $52.56 \pm 0.41$ | $52.56 \pm 0.41$ | $52.56 \pm 0.41$ | $52.56 \pm 0.41$ | $52.56 \pm 0.41$ | $52.56 \pm 0.41$ |
| $\alpha = 0$ | APPNP | $73.30 \pm 1.39$ | $76.14 \pm 1.07$ | $77.32 \pm 0.55$ | $76.42 \pm 0.65$ | $75.46 \pm 0.92$ | $70.38 \pm 1.03$ | $52.24 \pm 2.46$ |
| | $\mathrm{PMLP}_{APP}$ | $74.58 \pm 0.58$ | $77.08 \pm 0.77$ | $77.56 \pm 0.60$ | $76.68 \pm 0.77$ | $75.18 \pm 0.83$ | $71.00 \pm 0.85$ | $51.00 \pm 6.58$ |
| $\alpha = 0.1$ | APPNP+Res | $72.28 \pm 1.56$ | $74.82 \pm 1.31$ | $75.78 \pm 1.18$ | $76.18 \pm 1.11$ | $75.98 \pm 1.01$ | $76.04 \pm 0.92$ (Converged) | $76.04 \pm 0.92$ (Converged) |
| | $\mathrm{PMLP}_{APP+Res}$ | $72.96 \pm 0.75$ | $75.66 \pm 1.05$ | $76.68 \pm 0.82$ | $76.82 \pm 0.81$ | $76.86 \pm 0.87$ | $76.86 \pm 0.87$ | $76.86 \pm 0.87$ (Converged) |

Table 4: Layer number denotes the number of MP+FF layers, and the number of FF layers is fixed as 2. The hidden size is fixed as 64. For GCNII, $\mathbf{H}^{(\ell+1)} = \sigma\big(((1-\alpha_\ell)\,\mathrm{MP}(\mathbf{H}^{(\ell)}) + \alpha_\ell\mathbf{H}^{(0)})((1-\beta_\ell)\mathbf{I}_n + \beta_\ell\mathbf{W}^{(\ell)})\big)$, we set $\alpha_\ell = 0.1$, $\beta_\ell = 0.5/\ell$. ResNet here denotes MLP with residual connections (or equivalently, GCNII without MP operations). For JKNet, we use concatenation for layer aggregation. MLP+JK denotes MLP with jumping knowledge (or equivalently, JKNet without MP operations)

| | # Layers | 2 | 4 | 8 | 16 | 32 | 64 | 128 |
|---|---|---|---|---|---|---|---|---|
| w/o res. | MLP | $52.56 \pm 0.41$ | $48.18 \pm 4.17$ | $32.26 \pm 5.66$ | $30.64 \pm 6.75$ | $27.60 \pm 6.76$ | $19.86 \pm 3.25$ | $20.14 \pm 2.01$ |
| | GCN | $73.84 \pm 1.49$ | $74.86 \pm 3.03$ | $45.52 \pm 6.10$ | $33.50 \pm 5.23$ | $24.08 \pm 6.90$ | $27.82 \pm 2.76$ | $19.60 \pm 9.52$ |
| | $\mathrm{PMLP}_{GCN}$ | $73.96 \pm 0.78$ | $74.56 \pm 2.94$ | $44.98 \pm 6.51$ | $29.58 \pm 10.74$ | $26.66 \pm 5.31$ | $27.10 \pm 8.63$ | $27.76 \pm 7.14$ |
| with res. | ResNet | $53.08 \pm 1.15$ | $53.68 \pm 0.52$ | $53.12 \pm 1.57$ | $52.70 \pm 2.16$ | $49.36 \pm 1.70$ | $45.74 \pm 1.54$ | $40.74 \pm 2.39$ |
| | GCNII | $67.36 \pm 1.57$ | $74.54 \pm 1.02$ | $76.00 \pm 0.62$ | $76.28 \pm 0.68$ | $75.42 \pm 1.65$ | $75.18 \pm 0.87$ | $68.24 \pm 1.89$ |
| | $\mathrm{PMLP}_{GCNII}$ | $69.08 \pm 0.80$ | $74.98 \pm 0.51$ | $75.26 \pm 1.29$ | $75.62 \pm 1.48$ | $75.12 \pm 1.38$ | $72.64 \pm 2.24$ | $68.52 \pm 1.47$ |
| with res. | MLP+JK | $52.76 \pm 1.38$ | $53.46 \pm 1.29$ | $54.48 \pm 1.19$ | $56.10 \pm 0.73$ | $53.10 \pm 1.30$ | $49.38 \pm 1.96$ | $30.40 \pm 1.63$ |
| | JKNet | $65.50 \pm 0.78$ | $72.02 \pm 1.15$ | $72.22 \pm 1.08$ | $71.50 \pm 0.83$ | $71.38 \pm 1.61$ | $60.66 \pm 2.63$ | $51.52 \pm 4.00$ |
| | $\mathrm{PMLP}_{JKNet}$ | $67.82 \pm 0.78$ | $72.76 \pm 1.59$ | $74.02 \pm 0.53$ | $73.92 \pm 1.04$ | $72.76 \pm 1.28$ | $67.70 \pm 0.83$ | $44.70 \pm 6.47$ |

Table 5: Mean and STD of testing accuracy on datasets with high heterophily level.

| | Dataset | Chameleon | Squirrel | Film | Cornell | Texas | Wisconsin |
|---|---|---|---|---|---|---|---|
| GNNs | GCN | $56.27 \pm 1.60$ | $41.21 \pm 0.48$ | $29.88 \pm 0.56$ | $67.03 \pm 4.44$ | $63.24 \pm 8.02$ | $63.92 \pm 4.51$ |
| | SGC | $58.73 \pm 2.09$ | $43.73 \pm 1.95$ | $30.92 \pm 0.68$ | $58.92 \pm 10.54$ | $62.70 \pm 7.00$ | $66.67 \pm 4.60$ |
| | APPNP | $57.98 \pm 3.70$ | $41.59 \pm 1.50$ | $32.20 \pm 1.37$ | $65.95 \pm 3.08$ | $65.95 \pm 2.42$ | $66.27 \pm 2.56$ |
| MLPs | MLP | $49.61 \pm 1.36$ | $29.20 \pm 2.32$ | $36.43 \pm 2.21$ | $78.92 \pm 6.73$ | $76.76 \pm 3.08$ | $83.53 \pm 3.28$ |
| | $\mathbf{PMLP}_{GCN}$ $\Delta_{GNN}$ $\Delta_{MLP}$ | $57.02 \pm 3.74$ $+1.33\%$ $+14.94\%$ | $36.39 \pm 3.28$ $-11.70\%$ $+24.62\%$ | $29.75 \pm 1.16$ $-0.44\%$ $-18.34\%$ | $65.41 \pm 8.20$ $-2.42\%$ $-17.12\%$ | $62.70 \pm 11.04$ $-0.85\%$ $-18.32\%$ | $66.67 \pm 2.77$ $+4.30\%$ $-20.18\%$ |
| | $\mathbf{PMLP}_{SGC}$ $\Delta_{GNN}$ $\Delta_{MLP}$ | $55.83 \pm 2.51$ $-4.94\%$ $+12.54\%$ | $39.44 \pm 1.93$ $-9.81\%$ $+35.07\%$ | $28.66 \pm 0.61$ $-7.31\%$ $-21.33\%$ | $55.14 \pm 9.09$ $-6.42\%$ $-30.13\%$ | $62.70 \pm 4.44$ $0.00\%$ $-18.32\%$ | $62.35 \pm 5.26$ $-6.48\%$ $-25.36\%$ |
| | $\mathbf{PMLP}_{APP}$ $\Delta_{GNN}$ $\Delta_{MLP}$ | $57.28 \pm 2.25$ $-1.21\%$ $+15.46\%$ | $39.73 \pm 1.85$ $-4.47\%$ $+36.06\%$ | $31.54 \pm 0.74$ $-2.05\%$ $-13.42\%$ | $68.11 \pm 6.16$ $+3.28\%$ $-13.70\%$ | $65.95 \pm 8.24$ $0.00\%$ $-14.08\%$ | $67.45 \pm 4.51$ $+1.78\%$ $-19.25\%$ |

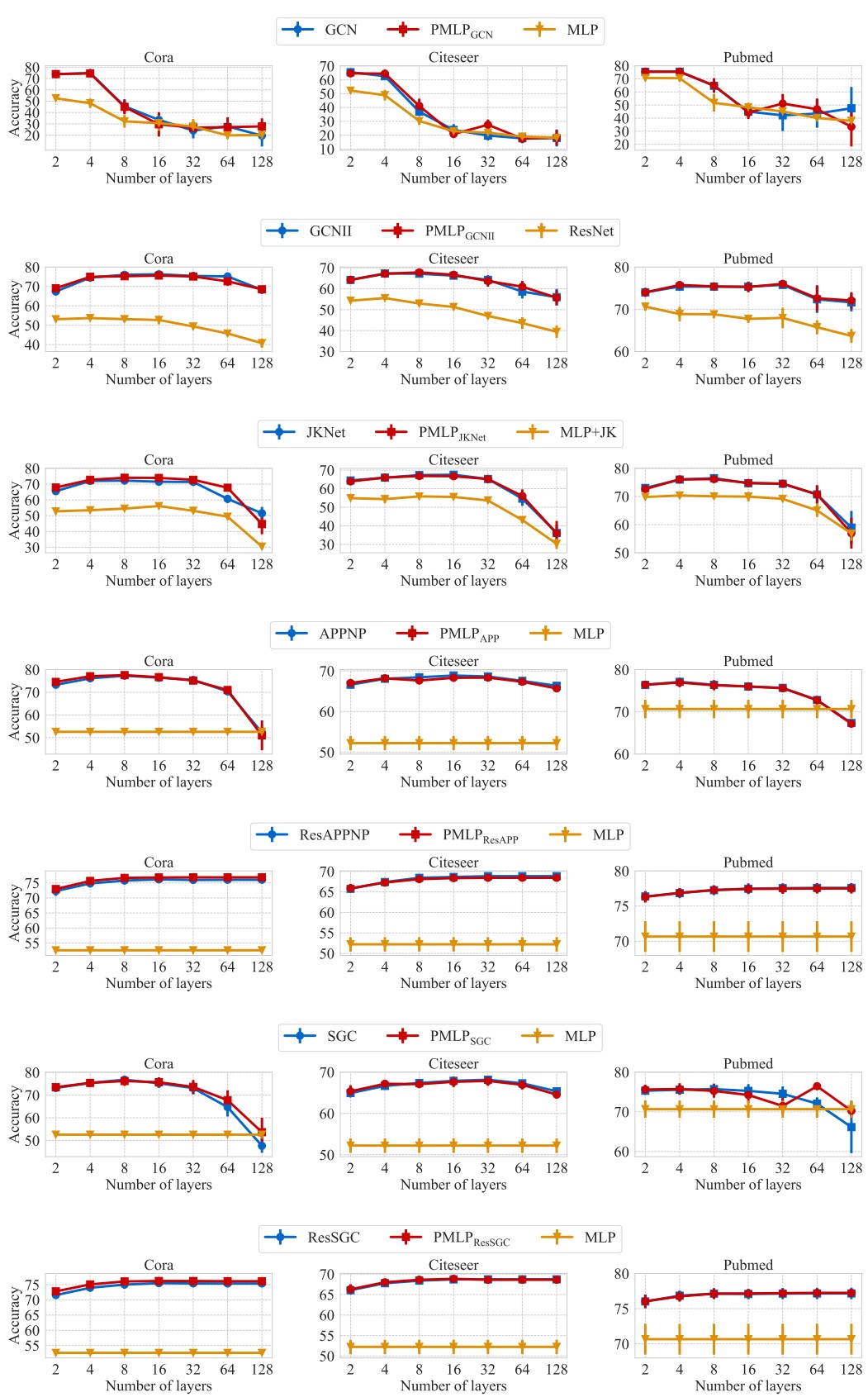

Figure 5: Performance variation of different GNN architectures with (GCNII, JKNet, ResAPPNP/SGC) or without (GCN, APPNP/SGC) residual links w.r.t. increasing layer number (from 2 to 128).

Table 6: Head-to-head comparison of three proposed PMLP models ($\text{PMLP}_{GCN}$, $\text{PMLP}_{SGC}$ and $\text{PMLP}_{APP}$) and the standard MLP.

| Model | Train and Valid | Inference | GNN Counterpart |
|---|---|---|---|
| MLP | | $\hat{y}_u = \psi(\mathbf{x}_u)$ | N/A |
| $\text{PMLP}_{GCN}$ | $\hat{y}_u = \psi(\mathbf{x}_u)$ | $\mathbf{h}_u^{(l)} = \psi^{(l)}(\text{MP}(\{\mathbf{h}_v^{(l-1)}\}_{v \in \mathcal{N}_u \cup \{u\}}))$ | GCN (Kipf & Welling, 2017) |
| $\text{PMLP}_{SGC}$ | | $\hat{y}_u = \psi(\text{Multi-MP}(\{\mathbf{x}_v\}_{v \in \mathcal{V}}))$ | SGC (Wu et al., 2019) |
| $\text{PMLP}_{APP}$ | | $\hat{y}_u = \text{Multi-MP}(\psi(\{\mathbf{x}_v\}_{v \in \mathcal{V}}))$ | APPNP (Klicpera et al., 2019) |

Table 7: Statistics of datasets.

| Dataset | # Classes | # Nodes | # Edges | # Node features |
|---|---|---|---|---|
| Cora (McCallum et al., 2000) | 7 | 2,708 | 5,278 | 1,433 |
| Citeseer (Sen et al., 2008) | 6 | 3,327 | 4,552 | 3,703 |
| Pubmed (Namata et al., 2012) | 3 | 19,717 | 44,324 | 500 |
| A-Photo (McAuley et al., 2015) | 8 | 7,650 | 119,081 | 745 |
| A-Computer | 10 | 13,752 | 245,861 | 767 |
| Coauthor-CS (Sinha et al., 2015) | 15 | 18,333 | 81,894 | 6,805 |
| Coauthor-Physics | 5 | 34,493 | 247,962 | 8,415 |
| OGBN-Arxiv (Hu et al., 2020) | 40 | 169,343 | 1,157,799 | 128 |
| OGBN-Products | 47 | 2,449,029 | 61,859,076 | 100 |
| Flickr (Zeng et al., 2019) | 7 | 89,250 | 449,878 | 500 |

# F  IMPLEMENTATION DETAILS

We present implementation details for our experiments for reproducibility. We implement our model as well as the baselines with Python 3.7, Pytorch 1.9.0 and Pytorch Geometric 1.7.2. All parameters are initialized with Xavier initialization procedure. We train the model by Adam optimizer. Most of the experiments are running with a NVIDIA 2080Ti with 11GB memory, except that for large-scale datasets we use a NVIDIA 3090 with 24GB memory. Table 6 summarizes the architecture of PMLPs adopted in training and testing stages for clear head-to-head comparison.

## F.1  DATASET DESCRIPTION

We use sixteen widely adopted node classification benchmarks involving different types of networks: three citations networks (Cora, Citeseer and Pubmed), two product co-occurrence networks (Amazon-Computer and Amazon-Photo), two coauthor-ship networks (Coauthor-CS, Coauthor-Computer), and three large-scale networks (OGBN-Arxiv, OGBN-Products and Flickr). For Cora, Citeseer and Pubmed, we use the provided split in (Kipf & Welling, 2017). For Amazon-Computer, Amazon-Photo, Coauthor-CS and Coauthor-Computer, we randomly sample 20 nodes from each class as labeled nodes, 30 nodes for validation and all other nodes for test following (Shchur et al., 2018). For two large-scale datasets, we follow the original splitting Hu et al. (2020) for evaluation. For Flickr, we use random split where the training and validation proportions are 10%. The statistics of these datasets are summarized in Table. 7.

## F.2  HYPERPARAMETER SEARCH

We use the same MLP architecture (i.e., number of FF layers and size of hidden states) as backbone for models in the same dataset, same GNN architecture (i.e., number of MP layers) for PMLP and its GNN counterpart, and finetune hyperparameters for each model including dropout rate (from $\{0.1, 0.2, 0.3, 0.4, 0.5, 0.6, 0.7, 0.8, 0.9\}$), weight decay factor (from $\{0, 0.0001, 0.001, 0.01, 0.1\}$), and learning rate (from $\{0.0001, 0.001, 0.01, 0.1\}$) using grid search.

For model architecture hyperparameters (i.e., layer number, size of hidden states), we fix them as reported in Table. 8, instead of fine-tuning them in favor of GNN or PMLP for each dataset which might introduce bias into their comparison. By default, we set (FF and MP) layer number as 2, and hidden size as 64, but manually adjust in case the performance of GNN is far from the optimal.

Table 8: Summary of model architectures.

| | | Cora | Citeseer | Pubmed | Photo | Computer | CS | Physics | Arxiv | Products | Flickr |
|---|---|---|---|---|---|---|---|---|---|---|---|
| **GCN** | Hidden Size | 64 | 64 | 64 | 32 | 256 | 128 | 128 | 64 | 64 | 64 |
| | # MP Layer | 2 | 2 | 2 | 2 | 2 | 2 | 2 | 2 | 2 | 2 |
| | # FF Layer | 2 | 2 | 2 | 2 | 2 | 2 | 2 | 2 | 2 | 2 |
| **PMLP$_{gcn}$** | Hidden Size | 64 | 64 | 64 | 32 | 256 | 128 | 128 | 64 | 64 | 64 |
| | # MP Layer | 2 | 2 | 2 | 2 | 2 | 2 | 2 | 2 | 2 | 2 |
| | # FF Layer | 2 | 2 | 2 | 2 | 2 | 2 | 2 | 2 | 2 | 2 |
| **SGC** | Hidden Size | 64 | 64 | 64 | 32 | 256 | 128 | 128 | 64 | 64 | 64 |
| | # MP Layer | 2 | 2 | 2 | 2 | 2 | 2 | 2 | 2 | 2 | 2 |
| | # FF Layer | 2 | 2 | 2 | 2 | 2 | 2 | 2 | 2 | 2 | 2 |
| **PMLP$_{sgc}$** | Hidden Size | 64 | 64 | 64 | 32 | 256 | 128 | 128 | 64 | 64 | 64 |
| | # MP Layer | 2 | 2 | 2 | 2 | 2 | 2 | 2 | 2 | 2 | 2 |
| | # FF Layer | 2 | 2 | 2 | 2 | 2 | 2 | 2 | 2 | 2 | 2 |
| **APPNP** | Hidden Size | 64 | 64 | 64 | 32 | 256 | 128 | 128 | 64 | 64 | 64 |
| | # MP Layer | 2 | 2 | 2 | 2 | 2 | 2 | 2 | 2 | 2 | 1 |
| | # FF Layer | 2 | 2 | 2 | 2 | 2 | 2 | 2 | 2 | 2 | 2 |
| **PMLP$_{app}$** | Hidden Size | 64 | 64 | 64 | 32 | 256 | 128 | 128 | 64 | 64 | 64 |
| | # MP Layer | 2 | 2 | 2 | 2 | 2 | 2 | 2 | 2 | 2 | 1 |
| | # FF Layer | 2 | 2 | 2 | 2 | 2 | 2 | 2 | 2 | 2 | 2 |
| **MLP** | Hidden Size | 64 | 64 | 64 | 32 | 256 | 128 | 128 | 64 | 64 | 64 |
| | # MP Layer | / | / | / | / | / | / | / | / | / | / |
| | # FF Layer | 2 | 2 | 2 | 2 | 2 | 2 | 2 | 2 | 2 | 2 |

Furthermore, we have discussed different settings for these architecture hyperparameters and find the performance of PMLP is consistently close to its GNN counterpart.

For other hyperparameters (i.e., learning rate, dropout rate, weight decay factor), we finetune them separately for each model on each dataset based on the performance on validation set. For PMLP, we use the MLP architecture for validation rather than using GNN since we find there is only slight difference in performance. In that sense, all PMLPs share the same training and validation process as the vanilla MLP, making them exactly the same model before inference.

# G    ADDITIONAL EXPERIMENTAL RESULTS

We supplement more experimental results in this section including extensions of the results in the main text and visualizations of the internal representations of nodes learned by 2-layer MLP, GNN, and PMLP on Cora and Citeseer datasets.

As we can see from Fig. 11-14, both PMLPs and GNNs show better capability for separating nodes of different classes than MLP in the internal layer, despite the fact that PMLPs share the same set of weights with MLP. Such results might indicate that the superior classification performance of GNNs mainly stem from the effects of message passing in inference, rather than GNNs' ability of learning better node representations.

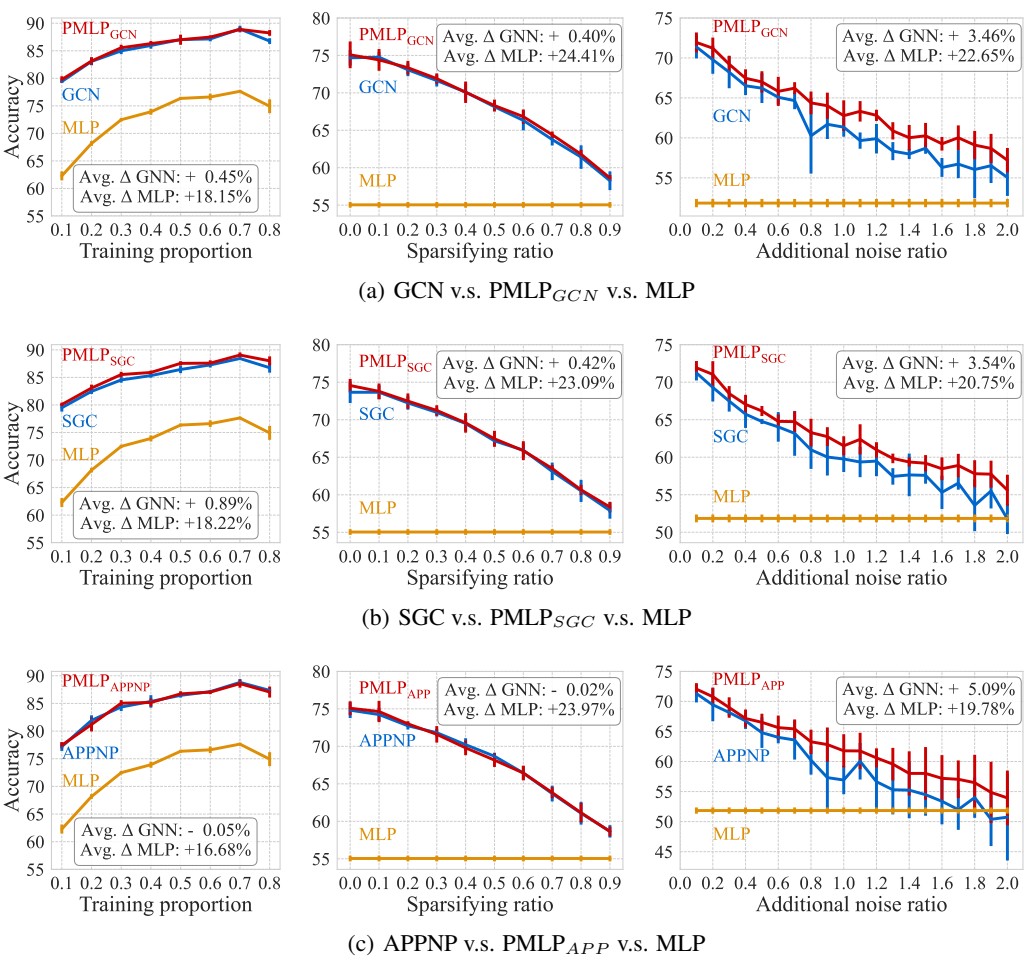

Figure 6: Impact of graph structural information by changing data split, sparsifying the graph, adding random structural noise on Cora.

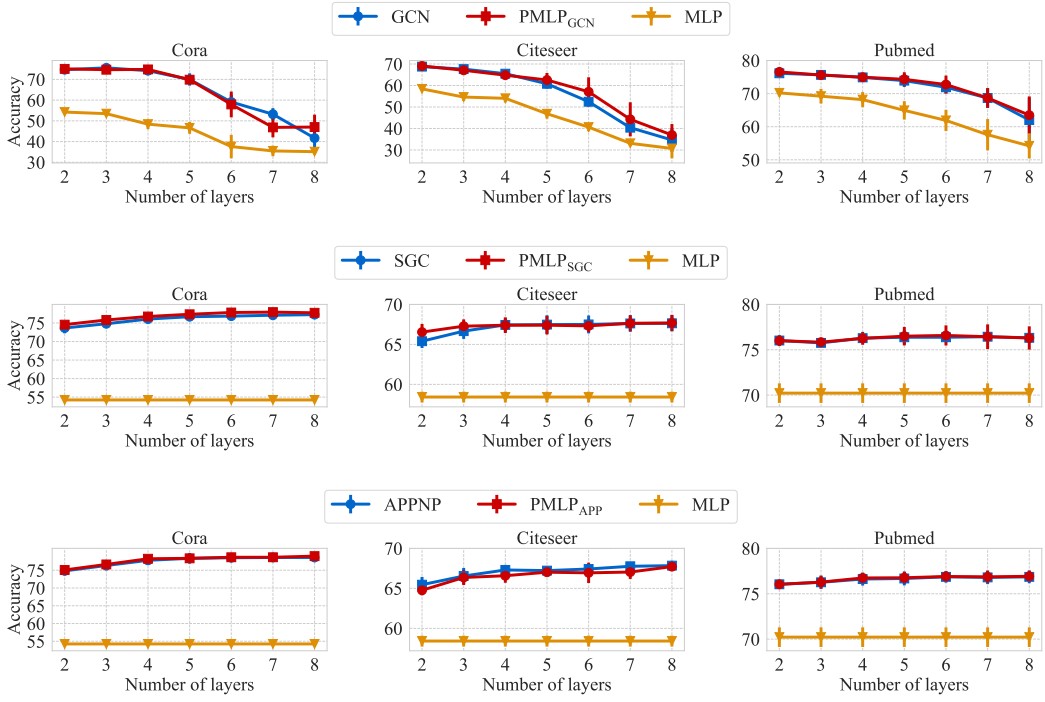

Figure 7: Performance variation with increasing layer number (from 2 to 8). Layer number here denotes number of FF and MP layers for GCN, and number of MP layers for SGC and APPNP.

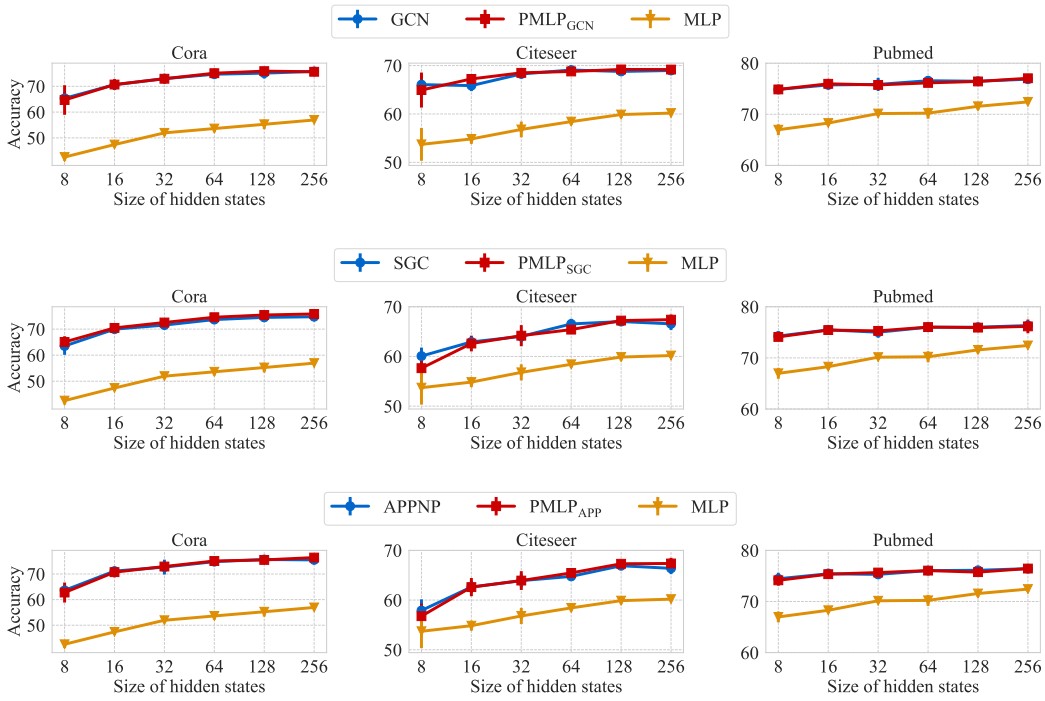

Figure 8: Performance variation with increasing size of hidden states.

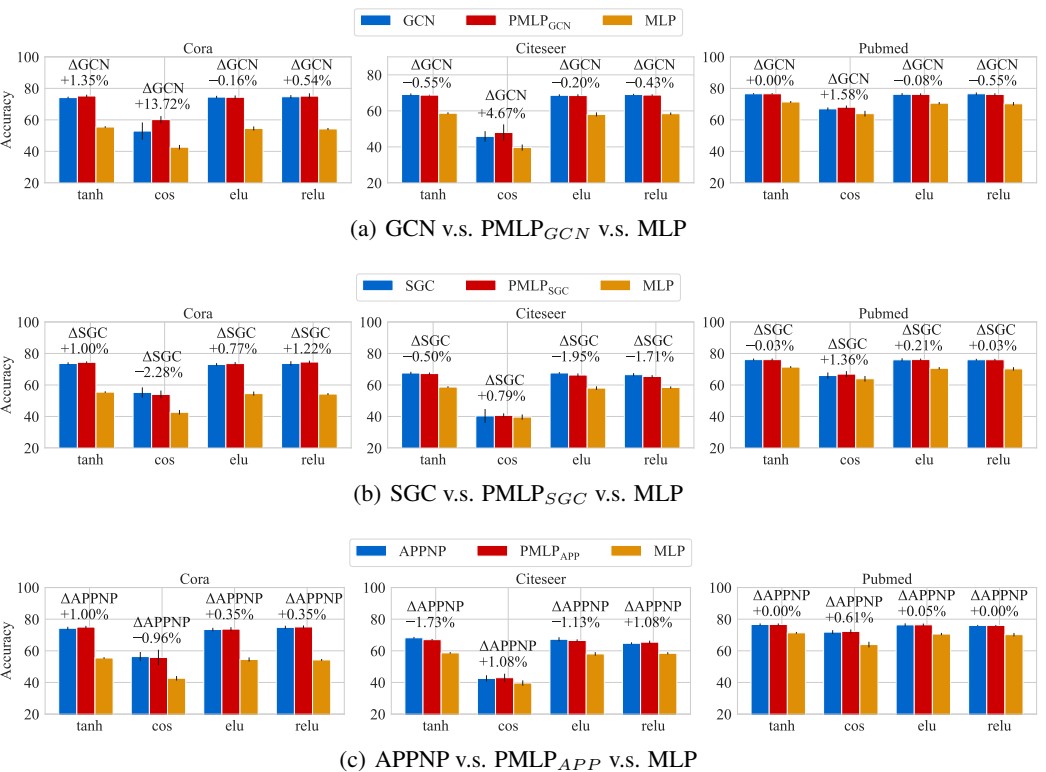

Figure 9: Performance variation with different activation functions in FF layer.

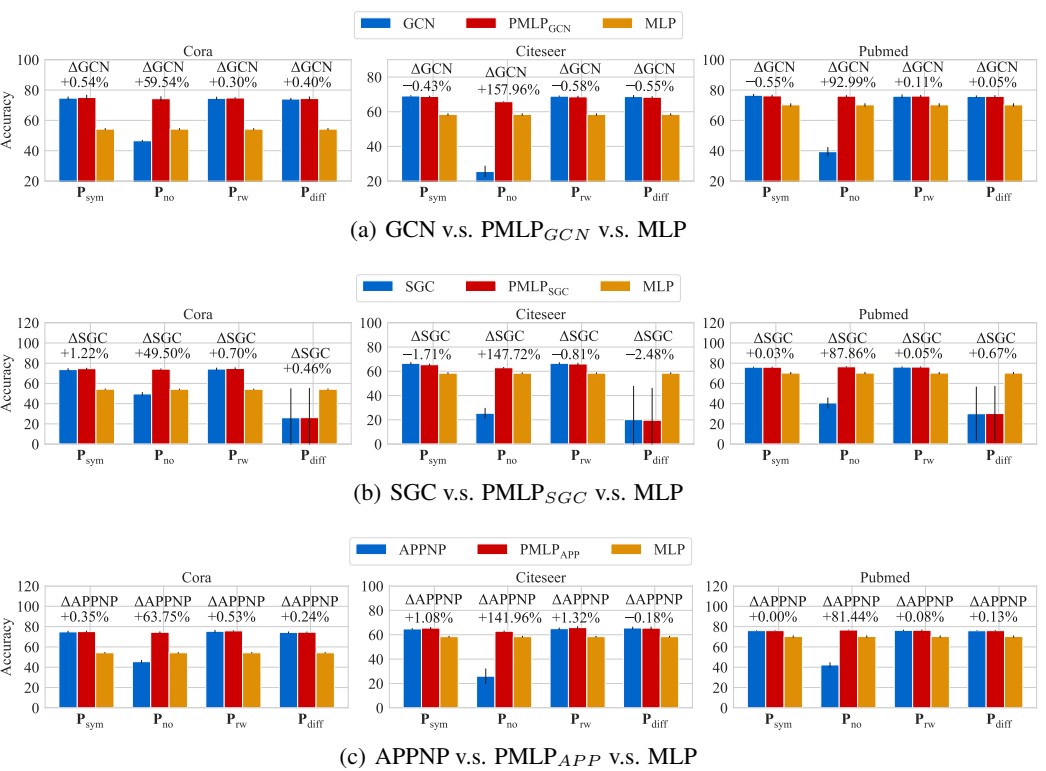

Figure 10: Performance variation with different transition matrices in MP layer.

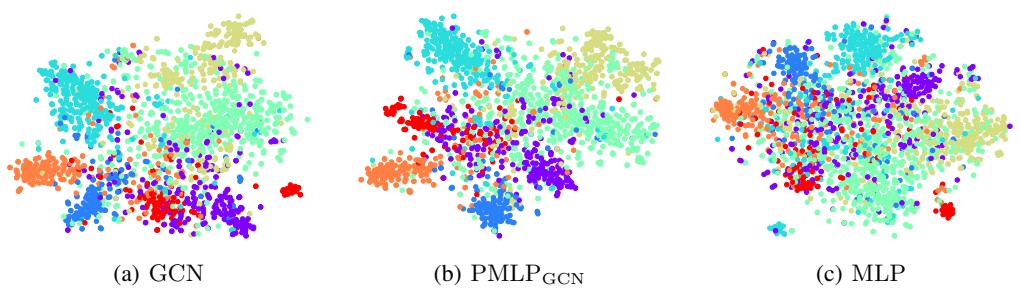

(a) GCN          (b) PMLP$_{GCN}$          (c) MLP

Figure 11: Visualization of node embeddings (2-D projection by t-SNE) in the internal layer for two-layer MLP, GCN and PMLP on Cora.

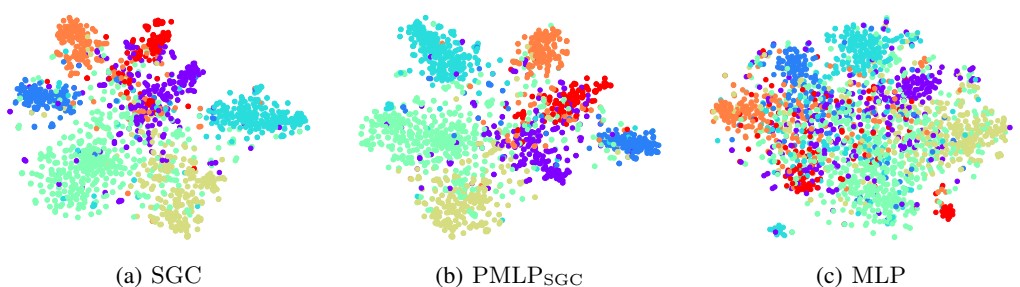

(a) SGC          (b) PMLP$_{SGC}$          (c) MLP

Figure 12: Visualization of node embeddings (2-D projection by t-SNE) in the internal layer for two-layer MLP, SGC and PMLP on Cora.

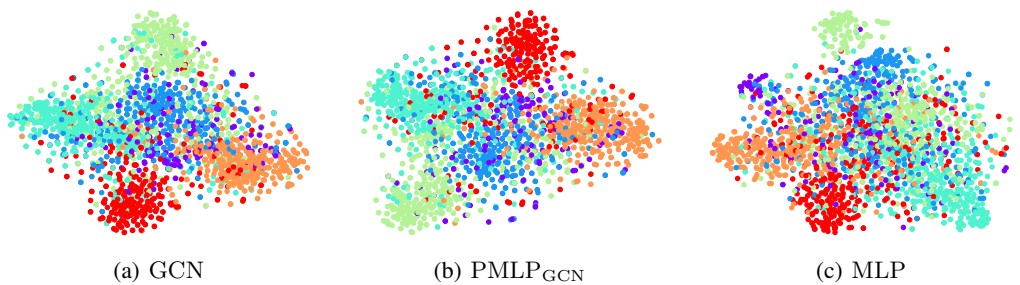

(a) GCN          (b) PMLP$_{GCN}$          (c) MLP

Figure 13: Visualization of node embeddings (2-D projection by t-SNE) in the internal layer for two-layer MLP, GCN and PMLP on Citeseer.

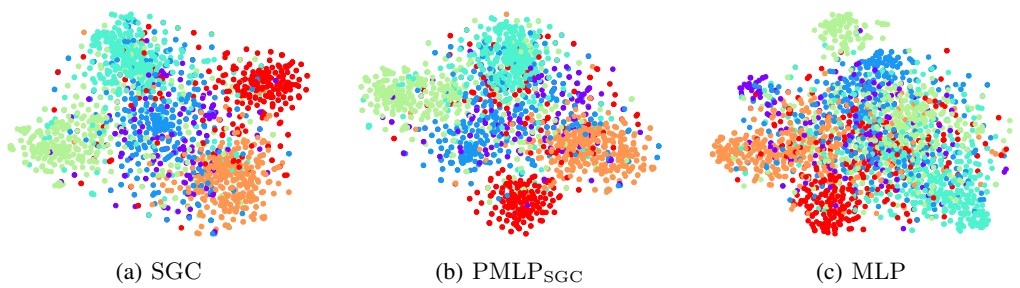

(a) SGC          (b) PMLP$_{SGC}$          (c) MLP

Figure 14: Visualization of node embeddings (2-D projection by t-SNE) in the internal layer for two-layer MLP, SGC and PMLP on Citeseer.

