# OpenReview forum: "Graph Neural Networks are Inherently Good Generalizers: Insights by Bridging GNNs and MLPs"
_ICLR.cc/2023/Conference — ICLR 2023 poster_

### Official Review · Reviewer_JEyU · 2022-10-24

**Confidence:** 4
**Correctness:** 4
**Technical Novelty And Significance:** 3
**Empirical Novelty And Significance:** 2
**Recommendation:** 6

**Clarity, Quality, Novelty And Reproducibility:**

The paper is sufficiently clear, and novel.
A few more information on the experimental settings are needed.
The paper presents a few grammar typos.
I found it strange that a citation to the original GNN work (Scarselli, Franco, et al. "The graph neural network model." IEEE transactions on neural networks 20.1 (2008): 61-80.) is missing in the work.


**Strength And Weaknesses:**

Strengths
* Novelty: the paper brings a new perspective in the analysis of GNN architectures (to the best of my knowledge)
* The mathematical analysis and the links to the NTK seem sound and interesting
* The experimental analysis seems comprehensive and relevant for the purposes of the paper

Weaknesses:
* Despite the paper is clear and the analysis interesting, I find surprising that the weights trained in the MLP work so well in most cases. Does this aspect mean that essentially to achieve a good classification it is sufficient to consider local nodes' information, and structural information encoding can be somehow neglected. It would be interesting to see a discussion on this aspect.
* Model selection and fine-tuning is not completely clear. In particular, it is not clear to me how the number of layers and neuron per layers are chosen, and if the hyper-parameters are tuned (on the validation set?) individually per each model and per task. Please clarify on this aspect for the sake of reproducibility of the results.


**Summary Of The Paper:**

The paper presents an analysis targeting the generalization in graph neural networks. The key idea is to decouple the node's information processing from the message passing operations. To this aim, it is introduced a new neural network paradigm, called PMLP, such that the weight values of a GNN are trained in the MLP form, and then only at inference stage the message passing operations are introduced. The approach is assessed in several tasks for node classification. To shed light on the reasons behind the good performances found in practice, the authors elaborate on the connections between MLP, PMLP, GNN and their respective neural tangent kernels equivalent.

Update after rebuttal: I thank the authors for addressing my concerns on the submitted version of the paper. I am happy to confirm my positive score.

**Summary Of The Review:**

In my view, the paper seems novel and interesting enough.
A few more details on the experimental settings (as specified above) can confirm the relevance of the achieved results.

---

> ### Author Response · Authors · 2022-11-15
> **Response to Reviewer JEyU**
>
> Thanks for the positive feedback and giving valuable comments for improvement. We hope the following responses will serve to address your concerns.
>
> > ***Q1: "Does this aspect mean that essentially to achieve a good classification it is sufficient to consider local nodes' information."***
>
> This statement is partially correct but not precise. The precise interpretation of our empirical finding is that to achieve good node classification requires the information of both local nodes and graph structure, yet the latter can be neglected in *most cases (but not all) during training (but not inference)*, which sheds new insights on some fundamantal aspects of GNNs' learning behaviors. See detailed illustration below.
>
> *(Empirical Findings)* Our main results reveal that, on most datasets, the PMLP model is consistently as effective as its GNN counterpart regardless of different implementations, architectures, hyper-parameters, etc. Still, as extensions of our experiments, we also find that on some large-scale datasets (Table.3), GNNs in general better fit the data and PMLPs perform slightly inferior than GNNs (but substantially outperform MLP). Another interesting finding is that the graph structure can even be harmful in training if it contains noisy edges (Fig.4,9).
>
> *(Implication and Significance)* Such phenomenon implies that when both GNNs and MLPs are expressive enough to fit the data, PMLP's superior performance can be attributed to the generalization ability of GNN architectures (used by PMLP in testing) and such advantage is inherent since it is not gained in training. This point is further supported by our theoretical analysis in Sec 4. Therefore, in most cases, local nodes' information is indeed sufficient for training a good node classifier. Nevertheless, there might also exist applications where the model expressivity plays an important role and MLP is insufficient for fitting the data (e.g., learning graph algorithms, classification on large datasets). In these cases and semi-supervised learning setting, incorporating graph structure during training  could be useful to further improve the testing accuracy.
>
>
> > ***Q2: ”It is not clear to me how the number of layers and neuron per layers are chosen, and if the hyper-parameters are tuned (on the validation set?) individually per each model and per task?”***
>
> For model architecture hyperparameters (i.e., layer number, size of hidden states), we fix them as reported in Table.5 in Appendix.D, instead of fine-tuning them in favor of GNN or PMLP for each dataset which might introduce bias into their comparison. By default, we set (FF and MP) layer number as 2, and hidden size as 64, but manually adjust in case the performance of GNN is far from the optimal. Furthermore, we have discussed different settings for these architecture hyperparameters and find the performance of PMLP is consistently close to its GNN counterpart as shown in Fig.2,5,6.
>
> For other hyper-parameters (i.e., learning rate, dropout rate, weight decay factor), we finetune them separately for each model on each dataset based on the performance on validation set. For PMLP, we use the MLP architecture for validation rather than GNN, but found that there is only slight difference in performance. In this sense, all PMLPs have the same training and validation process as the vanilla MLP, making them exactly the same model on the same dataset before inference. Despite the simple implementation of PMLP, codes will be made publicly available upon publication for reproducing the results. These detailed descriptions have been added in Appendix.D.2 in our revision.
>
> > ***Q3: ”A citation to the original GNN work is missing in the work.”***
>
> Thanks for pointing this out. Citations to [1] along with other classic works are added in our revision.
>
>
> [1] The graph neural network model, in IEEE transactions on neural networks 2008.

---

### Official Review · Reviewer_Hjsm · 2022-10-25

**Confidence:** 3
**Correctness:** 3
**Technical Novelty And Significance:** 3
**Empirical Novelty And Significance:** 3
**Recommendation:** 5

**Clarity, Quality, Novelty And Reproducibility:**

The paper is well written and easy to follow. But the proposed approach seems not a novel method.

**Strength And Weaknesses:**

**Pros:**

P1. Using PMLP which is much more efficient than ordinary GNNs achieves comparable results, and carry out extensive experiments.

**Cons:**

C1. The proposed methods (PMLP) is already used in APPNP. PMLP might not be a novel approach, and it could be just an extended version of APPNP from transduction to inductive setting.

C2. Since the performance of PMLP is similar to ordinary GCNs, the advantage of using PMLP is not significant. It would be better to specify much more persuasive property of the advantage of using PMLP.

C3. The propose theoretical results is far from the practical scenarios, and the claims in the theorem is ambiguous. Especially, I wonder it is effective to claim the extrapolation behavior of PMLP in inductive setting.

C4. In my knowledge, the authors does not specified how may ML operations are performed during the propagation steps. Because of the over-smoothing problem which is a well-known problem in GNN community, it would be better to specify the numbers. Furthermore, in my opinion, showing the results of using MP operations with residual connection would be important.

**Summary Of The Paper:**

This paper claims the statement that using message passing after propagating MLP can achieve comparable performance to using ordinary graph neural networks. By showing various empirical results, using P(ropagational) MLP and GNNs shows similar performance, and using PMLP is much more efficient than using GNNs. Furthermore, by proposing the theorems about the behavior of PMLP and MLP, authors claim that the ability of extrapolating the samples in PMLP can be treated as a useful property.

**Summary Of The Review:**

I vote to reject this paper because of the novelty of the proposed methods.

---

> ### Author Response · Authors · 2022-11-15
> **Response to Reviewer Hjsm (Part 1/2)**
>
> Thanks for your time and valuable comments that may help us for further improvement. We noticed that the reviewer has some lingering concerns on our novelty, which might be predicated on an assumption that our main contribution lies in the algorithmic novelty of PMLP. To resolve some potential misunderstandings that may affect how our work is interpreted, we'd like to clarify upfront that *the main aspect of our paper is NOT proposing a new approach with superiority over prior art, but instead, identifying a new empirical phenomenon that is pervasive across different GNN architectures, and revealing its profound implications and the underlying mechanism.* We next answer the raised questions with our contributions justified in specific.
>
> > ***Q1: "PMLP might not be a novel approach, and it could be just an extended version of APPNP from transduction to inductive setting."***
>
> Indeed, from the view of model architectures, the variant proposed for ablation study in the APPNP paper [1] can be roughly seen as a special case of PMLP (i.e., with APPNP-style architecture, personalized PageRank MP scheme, and residual connections). However, this does not weaken our main contributions since the empirical findings as well as the theoretical understandings, i.e., the central aspects of this paper, have not been identified before.
>
> The major contribution of [1] lies in the proposal of APPNP as a new *specific* GNN model and showing its empirical superiority over previous GNN models. In contrast, we introduce PMLP as a new *class* of model architectures that is applicable for a large array of GNNs, and more importantly, PMLPs are used for analysis purpose, based on which we identify a pervasive phenomenon that is consistent across MP instantiations, GNN architectures, hyper-parameters, etc. Beyond this new empirical finding, we also use PMLP as a lens to reveal that the main source of GNNs' success in node-level tasks is their inherent generalization ability brought by the MP operation used in inference, which is also justified by our theoretical analysis.
>
> As for the inductive setting, please kindly note that we were not to contrively be different from prior works that focus on the transductive setting, and instead, to achieve fair comparison (i.e., by using inductive setting to ensure that MLP, GNN and PMLP have access to the same training data information) and make our results reasonable and meaningful. To help make our big picture more clear in the presentation, we have added more discussions in Sec.1.1 to better position this work with prior art.
>
>
>
>
> > ***Q2: "Since the performance of PMLP is similar to ordinary GCNs, the advantage of using PMLP is not significant."***
>
> As mentioned above, our focus is not to show the superiority of PMLP over prior art, but use it as a general model class for empirical study and theoretical analysis on the fundamental aspects of how GNNs work. "The performance of PMLP is similar to ordinary GCNs" is exactly one of our key findings, i.e., the performance of a vanilla MLP can approach or even exceed a wide variety of GNNs regardless of their different architectural designs.
>
> The significance behind the identified phenomenon is centered at the fundamental understanding into the learning behaviors of GNNs and facilitating future theoretical research by bridging GNN and general neural networks. While the practical value of PMLP is a by-product, we do find it non-negligible, which includes but not limited in:
> - Substantial training efficiency and better robustness to noisy edges, as shown by our experiments.
> - Significantly speed up searching the optimal GNN architecture and hyper-parameters, by testing performance on top of a single pre-trained MLP (given the similar trend of performance variation between PMLP and GNN).
> - Great potential for further boosting the performance by incorporating the graph structural information that is originally neglected in training. E.g., we have some new results of PMLP combining with Label Propagation (LP) [2]:
>
> |  | GCN | MLP | PMLP$_{GCN}$ | PMLP$_{GCN}$+LP |
> | -------- | -------- | -------- | -------- | -------- |
> | Cora   | $74.82 \pm 1.09$ | $55.30 \pm 0.58$ | $75.86 \pm 0.93$ | $80.12 \pm 0.71$     |
> | Citeseer | $67.60 \pm 0.96$ | $56.20 \pm 1.27$ | $68.00 \pm 0.70$ | $71.76 \pm 0.86$|

---

> > ### Author Response · Authors · 2022-11-15
> > **Response to Reviewer Hjsm (Part 2/2)**
> >
> > > ***Q3: "The propose theoretical results is far from the practical scenarios, and the claims in the theorem is ambiguous. Especially, I wonder it is effective to claim the extrapolation behavior of PMLP in inductive setting."***
> >
> > Thank you for pointing out the potential ambiguity which helps us to improve the manuscript. However, our theory is well aligned with the practical setup and applicable for both inductive and transductive settings.
> >
> > In specific, our Theorem.4 indicates that when the testing node is distant from the training data range, both PMLP and GNN degrade to linear functions and fail to generalize (which is of theoretical interest, and indeed "far from the practical scenario"). Then, our Theorem.5 further implies when the testing node is near the training data range, PMLP and GNN can better extrapolate target functions. *Such affirmative case supports our empirical finding and aligns with the practical scenario.* As a real-world evidence, following [3], we find in Cora that only less than 1% of testing nodes are inside the convex hull of training data ($\mathcal H_{tr}$), 85.3% are outside $\mathcal H_{tr}$ but within distance $0.25r$ ($r$ is the diameter of $\mathcal H_{tr}$), and the rest are within distance $0.63r$ (different split yields similar results), namely most testing nodes are indeed "near the training data range".
> >
> > Furthermore, the analysis applies to both transductive and inductive settings since it focuses on the GNN structure itself ($\phi_{mlp}$ v.s. $\phi_{gnn}$) rather than effects of training. In specific, whether using unlabeled nodes only changes $\mathbf w_{gnn}^*$, which is not of interest in analysis and does not affect the conclusion. We have modified the remark below Thm.4 to clarify this point.
> >
> >
> > > ***Q4: "The authors does not specified how may MP operations are performed during the propagation steps. The results of using MP operations with residual connection would be important."***
> >
> > As has been specified in Table.5 in Appendix.D of the original draft, we use two layers of MP operation as the default setting rather than fine-tuning them in favor of GNN or PMLP which might introduce bias, and further discuss different choices of it (ranging from 2 to 8) in Fig.2,5,6. We have done additional experiments on over-smoothing and residual connection, with new results and insights that add to the novelty and significance of this work. Please refer to our general response titled "Model Depth, Over-Smoothing and Skip-Connection". More details and results are given in Appendix.E, some of which to be refined and put into the main text to further enrich our contributions. Thanks for this nice suggestion.
> >
> > [1] Predict then propagate: Graph neural networks meet personalized pagerank, in ICLR'19
> > [2] Label Propagation for Deep Semi-Supervised Learning, in CVPR'19
> > [3] Deep learning generalization and the convex hull of training sets, 2021

---

> ### Author Response · Authors · 2022-12-09
> **Look forward to feedback**
>
> Dear Reviewer Hjsm,
>
> Thanks for your time reviewing our paper! As we are approaching the end of the discussion phase, we'd like to know if our response has addressed your questions. If you have any further questions, we would be more than happy to address them. And if our response has addressed your concerns, we hope you could re-consider your evaluation in light of the revised paper and our initial response.
>
> Best regards,
> Authors

---

### Official Review · Reviewer_nT54 · 2022-10-27

**Confidence:** 4
**Correctness:** 3
**Technical Novelty And Significance:** 3
**Empirical Novelty And Significance:** 3
**Recommendation:** 8

**Clarity, Quality, Novelty And Reproducibility:**

## Clarify

Overall I think the paper is very well written with a good mixture of motivating examples, empirical results and theoretical insights. However, I suggest the authors make a careful pass on the proofs since I find some parts written unclearly (see above).

## Quality

I think the overall quality is high. The proposed model is novel, simple and efficient. The experiments are well designed. The theoretical analysis shows useful insights.

## Novelty

I think the novelty is high. The model itself demonstrates a new training and inference strategy. The theoretical analysis on generalization from the NTK perspective is also interesting.

## Reproducibility

The experimental setup and the hyperparameter searching procedures are described with sufficient details. Due to the simplicity of the model design, I think others can potentially reproduce the results.



**Strength And Weaknesses:**

## Strengths
+ The proposed is a simple model capturing some fundamental aspects of the learning process. The model computation process is very easy to understand and efficient to execute. Surprisingly, it achieves very promising empirical results. The fact that we can understand its extrapolation behavior under the lens of NTK is also very interesting.
+ The paper is novel. While there have been a few tricks in the literature to push GNNs towards MLPs by decoupling the message propagation from the feature transformation, most of them still require the message passing operation during training (e.g., by treating it as preprocessing). PMLP is different from them as the training is equivalent to a vanilla MLP. Theoretical analysis on generalization via NTK also brings some new aspects.
+ The paper is clearly written and I enjoy reading it. It shows a good mixture of empirical and theoretical studies.

## Weaknesses
- The proof can be written with more care. I have read most of the proof in detail. I do find some key steps hard to process during my first pass, due to missing details. I still have the following questions:
    - What does the superscript in $w^{(k)}$ mean? I suppose it denotes the dimension of $w$, and when $k\rightarrow \infty$, $w$ corresponds to an infinitely wide network. Is this correct?
    - What is the dimension of $\phi^{(1)}$? From the NTK definition, I think it should be a vector so that $(\phi^{(1)})^T \phi^{(1)}$ is a scalar. However, from Equation 33, it seems $\phi^{(1)}$ is a matrix (if $[\cdot,\cdot]$ denotes concatenation).
    - What does $...$ mean in the square brackets of Equation 33? In general, can you please clarify how to derive Equation 33 from Equation 30?
- It seems Theorem 4 and 5 apply to both PMLP and the original GNN. Therefore there is still a gap to explain why PMLPs achieve performance close to GNNs. Specifically, I think it is an important question why replacing the GNN weights with MLP weights does not hurt accuracy. This question is not addressed by Theorem 4 and 5 – basically they don’t care about what the weights look like.
- The setup of the theoretical analysis does not describe a realistic graph OOD scenario. When the test node $x_0$ is far away from the training nodes ($t\rightarrow \infty$), it likely won’t have any direct connections with the training nodes. i.e., it may have some multi-hop connections to the training nodes via other test nodes not too far away from the training nodes. In summary, I agree that the linear coefficient of extrapolation will depend on graph structure, but I don’t think $\tilde{d}\cdot \tilde{d_i}$ will be the realistic coefficient. Note that such concern does not apply to Theorem 3 on MLPs, since edges don’t exist in that case.


**Summary Of The Paper:**

This paper presents an efficient model, PMLP, which achieves the GNN-level generalization and accuracy at the cost of training an MLP (which is much cheaper than training a GNN). Specifically, the training of PMLP is identical to that of a normal MLP, where edge information in the graph is completely ignored. During inference, PMLP adds the message passing operation to the MLP model obtained during training. Thus, the inference of PMLP is similar to that of a normal GNN. The authors empirically show that PMLPs achieve similar test time accuracy as GNNs. PMLPs even achieve better robustness than GNNs when introducing structural noise to the graph. Theoretical analysis is also presented by analyzing the NTK of MLP, PMLP and GNN. It is shown that the message passing operation makes PMLPs and GNNs generalize differently from MLPs on out-of-distribution test data: while all models degrade to linear extrapolation on OOD data, PMLPs and GNNs have their linear coefficients depend on the node degree to preserve more information from the data.

**Summary Of The Review:**

Overall, I think this is a good paper. It is a simple model both revealing useful insights and enabling efficient practical deployment. The theoretical results seem correct (need some clarification from the authors). Yet there are some limitations of the analysis as I mentioned above. I vote for acceptance.

---

> ### Author Response · Authors · 2022-11-15
> **Response to Reviewer nT54**
>
> Thanks for appreciating our work and giving very detailed review. We hope the following responses will serve to address your concerns.
>
> > ***Q1: "What does the superscript in $w^{(k)}$ mean? / What does ... mean in the square brackets of Equation $33$? / What is the dimension of $\phi^{(1)}$? / How to derive Equation 33 from Equation 30?"***
>
> Here, $w^{(k)}\in \mathbb R^d$ is a random Gaussian vector whose dimension is fixed as $d$ (i.e. dimension of node features), with the superscript $^{(k)}$ denoting that it is the $k$-th sample among infinitely many i.i.d. sampled ones. Moreover, '...' in the square brackets of Eq.33 (Eq.35 in the new version) indicates that the two components repeat for infinitely many times with $k$ ranging from $1$ to $\infty$. Given that the first component in the square brackets of Eq.33 is in $\mathbb R^d$ and the second in $\mathbb R$, repeatly sampling $w^{(k)}$ and using concatenation yields an infinite-dimensional vector $\phi^{(1)} \in \mathbb R^{\infty}$.
>
> As for the connection between Eq.30 and Eq.33, computing expectation in Eq.30 corresponds to repeatly sampling $w^{(k)}$ and concatenating it with the previous results. The first/second term on the RHS of Eq.30 induces the first/second component in the square brackets of Eq.33 (by noting that $\boldsymbol{a}_i^{\top} \mathbf{G} \boldsymbol{a}_j=(\mathbf{X}^{\top} \boldsymbol{a}_i)^{\top} \mathbf{X}^{\top} \boldsymbol{a}_j$). Thank you for carefully checking our proofs and we have revised Lemma.2 and Appendix.B.3 with detailed mathematical derivations to improve the readability.
>
> > ***Q2: "It seems Theorem 4 and 5 apply to both PMLP and the original GNN. Therefore there is still a gap to explain why PMLPs achieve performance close to GNNs."***
>
> Indeed, the purpose of Theorem.4 and 5 is to shed theoretical insights on why "GNNs are inherently good generalizers" and explain our empirical finding that the superior generalizability of GNNs/PMLPs stems from the GNN architecture itself used in inference, rather than to strictly answer "why replacing the GNN weights with MLP weights does not hurt accuracy". The close performance between PMLPs and GNNs actually serves as empirical evidence that makes our theoretical results stronger: if weights were important for generalization, our theoretical results would be less convincing when "not caring about what the weights look like" in the analysis.
>
> That being said, we do agree it is important to understand the theoretical foundations behind the close performance between PMLPs and GNNs. As the first work revealing this phenomenon, we have also explored when PMLPs are superior or inferior than GNNs with corresponding discussions (Sec.3.2 and Sec.5), which involves noisy edges, self-connection, model expressivity, and role of unlabeled nodes. Incorporating all of them into analysis may suffice for another work in GNN theory. Therefore, we will leave this question for future research. We have modified the description at the beginning of Sec.4 and "Current Limitations and Outlooks" in Sec.5 to make this point more clear.
>
> > ***Q3: "The setup of the theoretical analysis does not describe a realistic graph OOD scenario."***
>
> The graph OOD scenario that the reviewer mentioned is actually compatible with our current analysis by changing the definition of $\mathbf X$ from features of training nodes to features of all nodes that are available in the dataset but not including $\boldsymbol x$ itself, and our final results will not be affected as long as $\mathbf X$ is fixed, which aligns with the real-world scenario. Nevertheless, there does exist a special scenario where the final linear coefficient will change accordingly: faetures of adjacent nodes in $\mathbf X$ also grow with $\boldsymbol x$ at similar rates as $t \rightarrow \infty$. However, such scenario lacks formal definition, and studying it might not be meaningful in our context, since the extreme case ($t \rightarrow \infty$) we considered is mainly of theoretical interest in order to characterize the function learned by GNNs, and both GNNs and PMLPs would fail to work in this extreme case regardless of different scenarios.

---

> > ### Comment · Reviewer_nT54 · 2022-11-21
> > **Updates from reviewer**
> >
> > I thank the authors for clarifying the theoretical analysis. I believe the rebuttal has addressed my main concerns, and I have raise the score. I think this paper provides some interesting perspectives in understanding the behaviors of GNNs.

---

### Author Response · Authors · 2022-11-15
**General Response**

We would like to thank the reviewers for their valuable feedback and detailed suggestions for improvement. Overall, the reviewers appreciate our work for that we reveal a "novel and surprising" (nT54, JEyU) phenomenon supported by extensive experiments" (Hjsm), paired with "sound and interesting" (JEyU) theoretical analysis and "useful insights" on "some fundamental aspects of GNN's learning process" (nT54). While some questions remain, we will address them in the response to each reviewer. The draft has been updated with modified parts colored blue.

---

> ### Author Response · Authors · 2022-11-15
> **Additional Experiments on Model Depth, Over-Smoothing, and Residual Connection**
>
> As part of the response to reviewer Hjsm, we have conducted experiments to shed additional lights on broader aspects of GNNs including model depth, over-smoothing and residual connection, which adds to the contributions of this work. In summary, we have the following new results and insights (see detailed comparison tables and companion discussions in our new Appendix E, part of which will be incorporated into the main text in the final version):
> 1. As shown by **new results in Table.6 and Table.7**, PMLP and its GNN counterpart (in GCN-, SGC- or APPNP-style) maintain close performance even for very deep (more than 100 layers) networks, despite that their (absolute) performance drops as layer increases, indicating over-smoothing is an issue for both PMLP and GNN without residual connections.
>
> 2. As we can see in Table.6, for GNNs using consecutive message passing layers (i.e. SGC-, APPNP-style), over-smoothing can be effectively addressed by 1) using message passing (MP) layers with residual connections in inference or even 2) only adding residual connections in inference if MP is already used in training. It is also worth mentioning that we need only to train a single MLP to fill all results of PMLPs in Table.6, and for layer number $128$, training MLP is $65/70$ times faster than SGC/APPNP.
>
> 3. As we can see in Table.7, for GCN-style GNNs whose MP layer and FF layer are coupled, training a MLP with residual connections similar to ResNet (or equivalently, GCNII [1] without message passing) and adding MP layers in inference yields a new model called $PMLP_{GCNII}$ whose performance is consistently close to the vanilla GCNII. Similar results are also observed between $PMLP_{JKNet}$ and JKNet [2].
>
> These results indicate that over-smoothing is more of a problem in inference and generalization rather than training, which is counter-intuitive but in alignment with [3] where the authors theoretically show very deep GNNs can still achieve high training accuracy but poor generalizability. Moreover, using residual connections (in inference for SGC- and APPNP-style GNNs, in both training and inference for GCN-style GNNs) might be an effective way to improve the model's generalization performance in case of over-smoothing.
>
> [1] Simple and Deep Graph Convolutional Networks, in ICML'20
> [2] Representation Learning on Graphs with Jumping Knowledge Networks, in ICML'18
> [3] On Provable Benefits of Depth in Training Graph Convolutional Networks, in NeurIPS'21

---

### Public Comment · ~Yihong_Luo1 · 2023-05-31
**Question about the Training time comparison**

Hi authors, thanks for your impressive work.

I have a question regarding your PeerMLP_SGC model. How is it possible for it to be 10 times faster than SGC?

As far as I know, SGC only propagates once and doesn't require any propagation during training. Therefore, the training time for SGC and MLP should be **the same**, except for the time needed to pre-compute the propagation.

Can you please explain the comparison of training time between the two models?

I really appreciate any response, thanks!

---

> ### Author Response · Authors · 2023-05-31
> **Thanks for your interest**
>
> In fact, to ensure consistent evaluation protocol across different GNN models (GCN, SGC, APPNP, GCNII, JKNet, etc.), we equally treat message passing operations in all models as part of their feed-forward computation in implementation, rather than being used for pre-processing. Our reported results are also based on such an implementational principle (see details in https://github.com/chr26195/PMLP). And of course in practice one can precompute node features (following [1-4]) for those GNNs where message passing operations are only used at the first layer. By adopting such implementational tricks, those particular GNN models should indeed have the same level of efficiency as their corresponding PMLP variants (not “PeerMLP”) as you mentioned. We will clarify this point later in our arxiv version. Should you have any further questions, please feel free to contact us via email (chr26195@sjtu.edu.cn).
>
> [1] Simplifying graph convolutional networks.
> [2] Are powerful graph neural nets necessary? a dissection on graph classification.
> [3] Sign: Scalable inception graph neural networks.
> [4] On graph neural networks versus graph-augmented mlps.

---

### Decision · Program_Chairs · 2023-01-20

**Decision:**

Accept: poster

**Justification For Why Not Higher Score:**

Modest novelty.

**Justification For Why Not Lower Score:**

The authors did a good job in responding to reviewers’ comments, leading to increased scores and an overall positive assessment from reviewers.

**Metareview: Summary, Strengths And Weaknesses:**

This paper presents an efficient model, PMLP, which achieves the GNN-level generalization and accuracy at the cost of training an MLP (which is much cheaper than training a GNN). The key idea is to decouple the node's information processing from the message passing operations. In PMLP, the weight values of a GNN are trained in the MLP form, and then only at inference stage the message passing operations are introduced. By showing various empirical results, using PMLP and GNNs shows similar performance, and using PMLP is much more efficient than using GNNs. PMLPs even achieve better robustness than GNNs when introducing structural noise to the graph. Theoretical analysis is also presented by analyzing the neural tangent kernels of MLP, PMLP and GNN. It is shown that the message passing operation makes PMLPs and GNNs generalize differently from MLPs on out-of-distribution test data: while all models degrade to linear extrapolation on OOD data, PMLPs and GNNs have their linear coefficients depend on the node degree to preserve more information from the data. The reviewers pointed to the gap to explain why PMLPs achieve performance close to GNNs, a realistic graph OOD scenario, differentiation with APPNP. The authors did a good job in responding to reviewers’ comments, leading to increased scores and an overall positive assessment from reviewers.

**Note From Pc:**

if the above contains the word "oral" or "spotlight" please see: "oral" presentation means -> notable-top-5% and "spotlight" means -> notable-top-25%. As stated in our emails, we are disassociating presentation type from AC recommendations